



# Opinion: The Scientific and Community-Building Roles of the Geoengineering Model Intercomparison Project (GeoMIP) - Past, Present, and Future

Daniele Visioni[1], Ben Kravitz[2,3], Alan Robock[4], Simone Tilmes[5], Jim Haywood[6,7], Olivier Boucher[8], Mark Lawrence[9], Peter Irvine[10], Ulrike Niemeier[11], Lili Xia[4], Gabriel Chiodo[12], Chris Lennard[13], Shingo Watanabe[14], John C. Moore[15,16,17], and Helene Muri[18]

[1]Sibley School of Mechanical and Aerospace Engineering, Cornell University, Ithaca, NY, USA
[2]Department of Earth and Atmospheric Science, Indiana University, Bloomington, IN, USA
[3]Atmospheric Sciences and Global Change Division, Pacific Northwest National Laboratory, Richland, WA, USA
[4]Department of Environmental Sciences, Rutgers University, New Brunswick, NJ, USA
[5]National Center for Atmospheric Research, Boulder, CO, USA
[6]Met Office Hadley Centre, Exeter, UK
[7]College of Engineering, Mathematics and Physical Sciences, University of Exeter, Exeter, UK
[8]Institut Pierre-Simon Laplace, Sorbonne Université/CNRS, Paris, France
[9]Institute for Advanced Sustainability Studies (IASS), Berliner Str. 130, 14467 Potsdam, Germany
[10]Earth Sciences, University College London, UK
[11]Max Planck Institute for Meteorology, Hamburg, Germany
[12]Institute for Atmospheric and Climate Science, ETH Zürich, Zürich, Switzerland
[13]Climate System Analysis Group, University of Cape Town, South Africa
[14]Japan Agency for Marine-Earth Science and Technology, Japan
[15]College of Global Change and Earth System Science, Beijing Normal University, Beijing, China
[16]CAS Center for Excellence in Tibetan Plateau Earth Sciences, Beijing, China
[17]Arctic Centre, University of Lapland, Rovaniemi, 96101, Finland
[18]Industrial Ecology Programme, Department of Energy and Process Engineering, Norwegian University of Science and Technology, Trondheim, Norway

**Correspondence:** Daniele Visioni (daniele.visioni@cornell.edu)

**Abstract.** The Geoengineering Model Intercomparison Project (GeoMIP) is a coordinating framework, started in 2010, that includes a series of standardized climate model experiments aimed at understanding the physical processes and projected impacts of solar geoengineering. Numerous experiments have been conducted, and numerous more have been proposed as "testbed" experiments, spanning a variety of geoengineering techniques aimed at modifying the planetary radiation budget: stratospheric
aerosol injection, marine cloud brightening, surface albedo modification, cirrus cloud thinning and sunshade mirrors. To date, more than one hundred studies have been published that used results from GeoMIP simulations. Here we provide a critical assessment of GeoMIP and its experiments.

We discuss its successes and missed opportunities, for instance in terms of which experiments elicited more interest from
the scientific community and which didn't, and the potential reasons why that happened. We also discuss the knowledge that GeoMIP has contributed to the field of geoengineering research and climate science as a whole: what have we learned in terms





of inter-model differences, robustness of the projected outcomes for specific geoengineering methods and future areas of models' development that would be necessary in the future. We also offer multiple examples of cases where GeoMIP experiments were fundamental for international assessments of climate change.

Finally, we provide a series of recommendations, regarding both future experiments and more general activities, with the goal of continuously deepening our understanding of the effects of potential geoengineering approaches, as well as reducing uncertainties in climate outcomes, important for assessing wider impacts on societies and ecosystems. In doing so, we refine the purpose of GeoMIP and outline a series of criteria whereby GeoMIP can best serve its participants, stakeholders, and the
broader science community.

## 1   Introduction

The comparison of results from nominally identical experiments in multiple, distinct climate models can be a very useful tool for understanding models' biases, robustness in the climate response to external forcings, and for partitioning sources of uncertainties in future climate projections Lehner et al. (2020). There is a long history of these model intercomparison projects
(MIPs) going back several decades Cess et al. (1989). This process has become more formalized and rigorous and now falls under the auspices of the Coupled Model Intercomparison Project (CMIP; (Meehl et al., 2005)), which is one of the flagship efforts of the World Climate Research Programme. CMIP is key to our understanding of future climate change projections, and its results are prominently featured in the Intergovernmental Panel on Climate Change's assessment reports, among other numerous studies. With Phase 6 (CMIP6; (Eyring et al., 2016)), the decision was made to move from a centralized effort to
a more distributed MIP approach, allowing different modeling groups to focus on different aspects of the Earth system. One of the most widely used satellite MIPs is ScenarioMIP, which aims to produce "multi-model climate projections based on alternative scenarios of future emissions" (O'Neill et al., 2016), forming the basis for future projections of climate change. There are over 20 other MIPs spanning a wide variety of research topics, including the Chemistry-Climate Model Initiative Morgenstern et al. (2017), aimed at evaluating models projections of the stratospheric ozone layer, tropospheric composition,
and interactions with climate, the Volcanic Forcing Model Intercomparison Project (VolMIP, Zanchettin et al. (2016)) aimed at assessing the robustness of the modeled response of the atmospheric-oceanic coupled system to a volcanic forcing, and the Land Model Intercomparison Project (LUMIP, Lawrence et al. (2016)), aimed at understanding the climatic contribution of changes in land-use activities.

Global mean surface air temperature in the decade 2011-2020 is around 1.1°C higher than pre-industrial period (Chen et al., 2021), and most future climate projections suggest continued warming in the future, with only a very few ambitious scenarios managing to stabilize temperatures in the second half of the century. The rate of warming in recent decades is unprecedented in at least the last 2000 years. Mitigation efforts to reduce emissions of greenhouse gases, that are the root cause of global warming and the associated climate change have, so far, been largely insufficient, with concentrations of atmospheric carbon



dioxide over the last thirty years being accurately represented by the IS92a 'business as usual' scenario that was developed over thirty years ago. Even though countries across the world agreed in the Paris Agreement (UNFCCC, 2015 [1]) to keep "the increase in global average temperature to well below 2°C above pre-industrial levels and pursuing efforts to limit temperature increase to 1.5 °C", their submitted Intended Nationally Determined Contributions (INDCs) of greenhouse gas emissions would be consistent with a projected median warming of between 2.6–3.1 °C by 2100 (Rogelj et al., 2016).


    Recognising these facts, around 10 years ago, an international group of researchers (Kravitz et al., 2011) proposed a new framework to coordinate climate modeling experiments to study proposals for solar geoengineering (also known as Solar Radiation Modification or Climate Intervention), aimed at understanding the impacts of proposed methods to offset the warming produced by an increase in greenhouse gases by directly intervening in the Earth's radiative balance. Fundamentally, these

studies aim to produce a negative radiative forcing by targeting the planetary albedo to partly counteract the positive forcing of $CO_2$ and other greenhouse gases (for a comprehensive review of the scientific aspects raised by geoengineering techniques, see for instance Lawrence et al. (2018), Kravitz and MacMartin (2020) and international reports such as those from the National Academy of Science, Engineering and Medicine, of Sciences Engineering and Medicine (2021) and EuTRACE, Aaheim et al. (2015). The issue of solar geoengineering had already been discussed in the past (see for instance Budyko (1978), Keith (2000)

and Govindasamy et al. (2003)), but the 2006 Editorial Essay by Paul Crutzen "Albedo Enhancement by Stratospheric Sulfur Injections: A Contribution to Resolve a Policy Dilemma?" (Crutzen, 2006) perhaps contributed more than anything else in drawing the attention of the scientific community on the topic, as can be seen by the immediate responses it elicited (Bengtsson, 2006; Cicerone, 2006; Kiehl, 2006; MacCracken, 2006).

This international framework, the **Geoengineering Model Intercomparison Project (GeoMIP)**, was initially coordinated with parallel work in an European Union project named Implications and risks of engineering solar radiation to limit climate change (IMPLICC), which included the intercomparison of simulations of four climate models for some of the same simulation setups as used in the first round of GeoMIP simulations (Schmidt et al., 2012). The motivation for the initiation of this project was the lack of consistency between initial geoengineering studies, which resulted in very different climate outcomes,

complicating the process of disentangling some of the observed differences (Rasch et al. (2009); Jones et al. (2009)). A set of standardized experiments comprising a reduction in the solar constant and the injection of $SO_2$ in the equatorial stratosphere was proposed and later expanded Kravitz et al. (2013a, 2015) to encompass other geoengineering techniques (such as marine cloud brightening and cirrus thinning), and following climate change scenarios described by CMIP6. The GeoMIP community has produced over 100 papers (121 in November 2022 following the self-reported list tracked on the GeoMIP website [2]) dis-

cussing or analyzing the impacts of these standardized experiments on the atmosphere, ocean, ecosystems and human societies. As an officially endorsed part of CMIP, GeoMIP has enjoyed a collaborative relationship with other MIPs, exchanging findings and lessons learned, as well as co-developing experiment protocols, definitely moving our knowledge of climate geoengineer-

---

[1]The Paris Agreement, https://unfccc.int/documents/184656

[2]http://climate.envsci.rutgers.edu/GeoMIP/publications.html, last accessed November 3rd, 2022





ing forward compared to 10 years ago.

The purpose of this paper is to take stock of GeoMIP as a project based on our collective experience in this field. What have been its successes in advancing knowledge around geoengineering or climate science as a whole? What are some shortcomings with its experiments, analysis, or coordination? What collaborations has it facilitated, and what are some missed opportunities? And, perhaps most importantly, what are its next steps and the outstanding questions it needs to address? There are, of course, few objective answers to these questions; we have identified when we are making subjective judgments and upon what values those opinions are based. In the next three sections we provide overviews of past and present GeoMIP experiments, testbeds for experiment development, and planned future experiments. Following that, in Section 5, we reflect on the role of GeoMIP, and provide our conclusions and outlook in Section 6.

## 2 An assessment of past and present GeoMIP experiments

The CMIP protocol specifies that MIP experiments be divided into tiers. Tier 1 experiments are the highest priority and are sometimes considered the minimum requirements for participation in that MIP. Subsequent tiers, while scientifically relevant, are considered lower priority. The philosophy of GeoMIP has always been to keep the number of Tier 1 experiments small so as to reduce barriers to participation and increase the number of models conducting these core experiments. In Table 1, we provide a summary of all the formally adopted GeoMIP experiments to date, including the number of models that have participated in each experiment. Tier 1 experiments are summarized in Figure 1.

### 2.1 Solar dimming: G1, G1ext, G2

Of all possible experiments, the simplest and easiest to replicate in different climate models, aims to offset the radiative forcing from an increase in $CO_2$ with a reduction in the model's solar constant. This method directly represents the idea of space sunshades (Angel, 2006): while potentially effective, technical feasibility and costs associated with deployment make such an approach prohibitive when compared to other proposed climate geoengineering methods (e.g. (The Royal Society (London), 2009)). More relevant for immediately practical geoengineering methods, solar dimming approximates the broad radiative effects of stratospheric aerosol injection.

Experiments G1 (CMIP5) and G1ext (CMIP6) involved offsetting the forcing from an instantaneous quadrupling of the $CO_2$ concentration (abrupt4xCO2, a standard CMIP experiment) with solar constant reduction. The high level of replicability means that results between CMIP5 and CMIP6 could be easily compared (Kravitz et al., 2021), even in some cases allowing for a comparison between different model versions; in Fig. 2 we show such a comparison across CMIP5 and CMIP6 models reproduced from Kravitz et al. (2021). G1 has been extensively studied in terms of the hydrological response (Tilmes et al., 2013; Kravitz et al., 2013b, 2014) and from an energetic and thermodynamics perspective (Russotto and Ackerman, 2018a, b;





**Table 1.** Summary of all experiments in GeoMIP, with the specific reference of the paper in which they were described for further details (last column). CDNC = cloud droplet number concentration, GHG = greenhouse gases, ODS = ozone-depleting substance, PI = preindustrial, SST = sea surface temperature.

| Experiment name | Description | Participating models | Tier | First described in |
|---|---|---|---|---|
| G1 + G1ext | Solar constant reduction to counteract $4xCO_2$ increase | 13 (G1) 7 (G1ext) | 1 | Kravitz et al. (2011) Kravitz et al. (2015) |
| G1ocean-albedo | Ocean albedo increase to counteract $4xCO_2$ increase | 12 | 1 | Kravitz et al. (2013a) |
| G2 | Balance 1% $CO_2$ increase per year via solar irradiance reduction | 11 | 1 | Kravitz et al. (2011) |
| G3 | $SO_2$ injections to counteract increasing GHG forcing from RCP4.5 | 7 | 1 | Kravitz et al. (2011) |
| G3S | Solar dimming to counteract increasing GHG forcing from RCP4.5 | 1 | 1 | Niemeier et al. (2013) |
| G3-SSCE | Sea salt injections to counteract increasing GHG forcing from RCP4.5 | 3 | 1 | Alterskjaer et al. (2013) |
| G4 | Constant $SO_2$ injections on RCP4.5 background | 10 | 1 | Kravitz et al. (2011) |
| G4cdnc | 50% increase in the CDNC of marine low clouds on RCP4.5 background | 9 | 1 | Kravitz et al. (2013a) |
| G4sea-salt | Sea spray intervention to offset a fixed amount of top of atmosphere forcing on RCP4.5 background | 3 | 1 | Kravitz et al. (2013a) |
| G4foam | Localized increase in ocean albedo on RCP6.0 background | 1 | Testbed | Gabriel et al. (2017) |
| G4SSA | Specified aerosol field for Climate-Chemistry models | 1 | Testbed | Tilmes et al. (2015) |
| senD2-sai | Specified aerosol field for CCMI-2022 | 4 | Testbed | |
| G6solar | Solar reduction to reduce increasing temperatures from SSP5-8.5 to SSP2-4.5 | 6 | 1 | Kravitz et al. (2015) |
| G6sulfur | $SO_2$ injections to reduce increasing temperatures from SSP5-8.5 to SSP2-4.5 | 6 | 1 | Kravitz et al. (2015) |
| G7cirrus | Reduce cirrus cloud optical depth by a constant amount on SSP5-8.5 | 2 | 2 | Kravitz et al. (2015) |
| Overshoot | $SO_2$ injections to keep temperatures at 1.5 and 2 °C above PI in SSP5-8.5 and SSP5-3.4OS | 1 | Testbed | Tilmes et al. (2020) |
| $H_2SO_4$ | Fixed $SO_2$ and $H_2SO_4$ injections, 2040 conditions for ODS and GHG, fixed SSTs at 1990 levels. | 3 | Testbed | Weisenstein et al. (2021) |



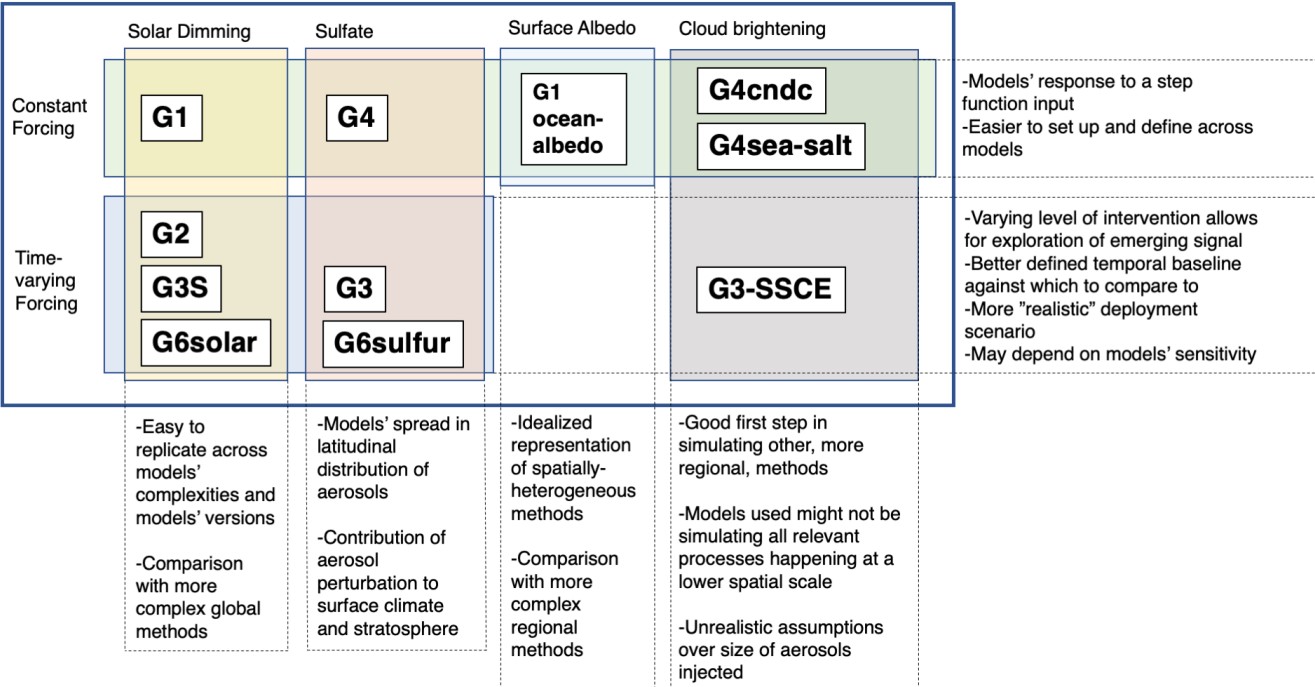

**Figure 1.** Schematic summary of all Tier-1 GeoMIP experiments across different iterations (Kravitz et al., 2011, 2013a, 2015). In rows, experiments are categorized based on how the geoengineering forcing is applied: constant or time-varying. In columns, experiments are categorized based on the method of applied forcing.

Virgin and Fletcher, 2022), highlighting both some commonalities in the response of the hydrological cycle to a reduction in incoming shortwave radiation, but also some large discrepancies in the cloud response. Similar experiments have also been performed outside of the GeoMIP framework: for instance, Irvine et al. (2019) used a higher resolution model to understand changes in extremes and precipitation in a case where a doubling of preindustrial $CO_2$ is partially offset by a reduction in solar constant. Results from G1 have been used for more specific impact analyses (e.g., Bal et al. (2019); Harding et al. (2020)),

which poses two important issues. First, G1 is an extreme, idealized case, and thus a straightforward analysis of climate model output from G1 cannot be used as a prediction of what climate engineering would do under any practical deployment strategy. This is particularly true for variables such as regional precipitation for which solar dimming is a poor proxy for aerosol injections or other methods (see Niemeier et al. (2013); Visioni et al. (2021a)) or for comparisons of tropical and high latitude effects, considering that uniformly reducing the solar constant tends to produce a stronger cooling in the tropics (e.g.,

Govindasamy et al. (2003); Kravitz et al. (2013b)). Second, there is no single answer for what climate engineering "would do," as the effects of climate engineering can, to some degree, be designed to mitigate any residual climate change impacts (e.g., Kravitz et al. (2016)); this is discussed further in Section 3.5 below.



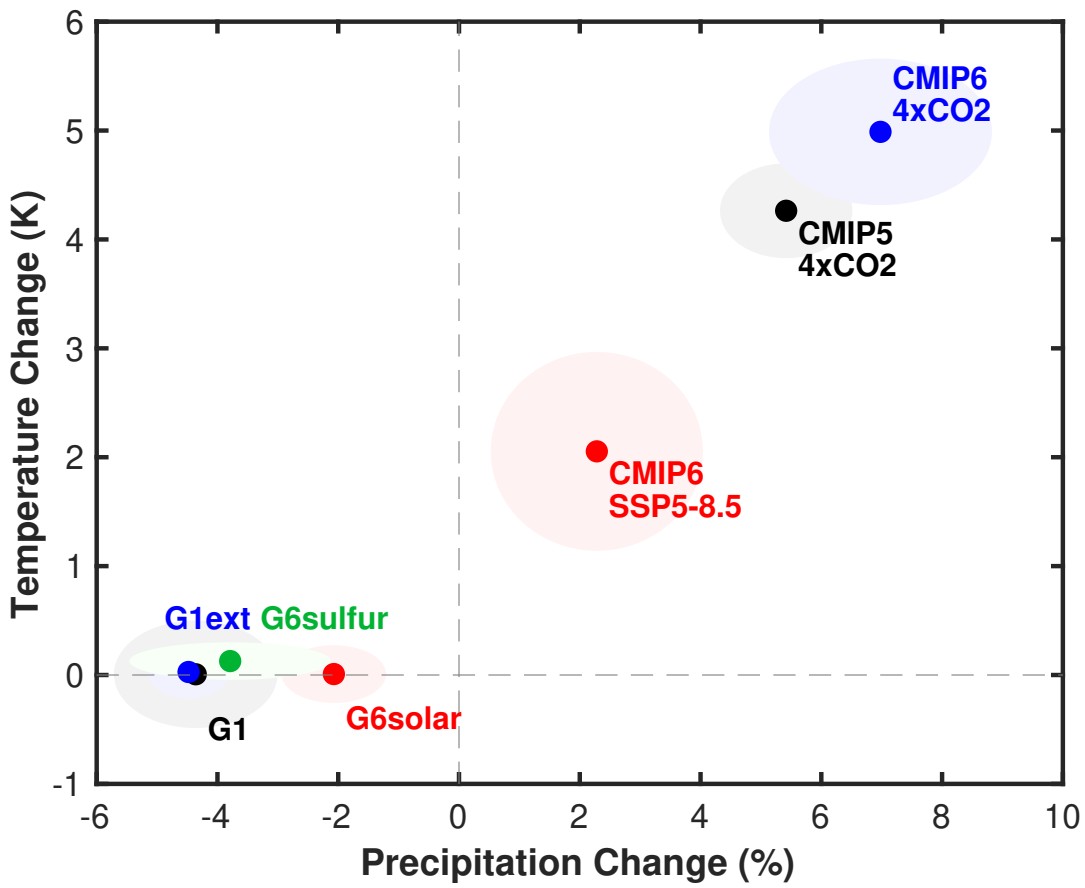

**Figure 2.** A comparison of global temperature (K) and precipitation (%) changes for some Tier 1 GeoMIP experiments across CMIP5 and CMIP6. Points represent the multi-model averages for each experiment, shaded areas represent 2 multi-model standard deviations. Values for G1 and 4xCO2 (CMIP5, 13 models averaged) and G1ext and 4xCO2 (7 models) are from Kravitz et al. (2021)), comparing against piControl values. Values for G6solar, G6sulfur and SSP5-8.5 (6 models) are from Visioni et al. (2021b), comparing against SSP2-4.5 values.

G2 similarly prescribes a solar reduction to offset an increase in the CO2 concentration, but in this case CO2 is increased by 1% every year, and the solar constant is successively reduced each year. G2 has been more seldom used, and usually as a way to test linearity assumptions in G1 (MacMartin and Kravitz, 2016). It was, however, featured in the first multi-model intercomparison of the termination effect (Jones et al., 2013); the gradual change in forcing allowed for a computation of rates of climate change under geoengineering with rates of change under termination.

G1 has substantial advantages that merit keeping it in future iterations of GeoMIP. First, while solar dimming does not capture some important features of climate response to stratospheric aerosol injection (Visioni et al., 2021a), it does capture some of the broad radiative effects, giving an indication of some of the radiative impacts. Also, G1 is easy to perform in all climate models, providing a low barrier to participation in GeoMIP, which is important for community-building and developing





high confidence in results. Nevertheless, its limits as compared to more detailed representations of the effects of more practical geoengineering methods should be always well communicated (Reynolds, 2022).

## 2.2 Surface albedo modification: G1ocean-albedo

Marine cloud brightening (MCB; Latham (1990)) is also a commonly researched method of conducting geoengineering, however simulating MCB in a multi-model context has proven challenging for experimental design. Pre-GeoMIP simulations have either injected sea salt aerosols directly into the marine boundary layer (e.g. Jones and Haywood (2012)) or increased the cloud droplet number concentration (CDNC) in marine low clouds (e.g. Jones et al. (2009); Rasch et al. (2009)). However,

because different models have different cloud cover amounts and locations, any multi-model comparison of these methods will necessarily impose different amounts and locations of forcing. While this can still be useful for a multi-model comparison (see Section 2.5 below), a more idealized experiment with a more controlled forcing could also be useful.

G1ocean-albedo involves offsetting the forcing from an abrupt quadrupling of the $CO_2$ concentration with an increase in surface albedo over all ocean regions (Kravitz et al., 2013a). While only loosely approximating the effects of MCB, it does

capture differential forcing between land and ocean, as well as a different perturbation to column absorption and vertical motion than would result from solar dimming. One study thus far has looked at a multi-model comparison of G1ocean-albedo results, finding that even though the models were in net top-of-atmosphere energy balance, global average temperature increased due to differential warming of the atmosphere and ocean, resulting in increased land-ocean energy transport (Kravitz et al., 2018). While perhaps this experiment has limited relevance for understanding potential geoengineering deployments, it does indicate

the usefulness of geoengineering simulations for understanding fundamental climate responses to forcing.

## 2.3 Stratospheric aerosol injections: G3 and G4

One of the first proposed methods of conducting geoengineering is to mimic the cooling effects of a volcanic eruption by injecting sulfate aerosol precursors into the (tropical) stratosphere (Budyko, 1978). The first general circulation model simulations of this method of geoengineering (Robock et al., 2008) described their simulations in terms of $\frac{1}{4}$ or $\frac{1}{2}$ of a Pinatubo eruption

every year (5 Tg $SO_2$ and 10 Tg $SO_2$ per year, respectively, based on some Pinatubo estimates, Timmreck et al. (2018)). The experiments G3 and G4 Kravitz et al. (2011) aimed to reproduce this in multiple models: in G3, the injection of sulfate was aimed at maintaining net top-of-atmosphere radiative forcing at 2020 levels under an RCP4.5 scenario, and in G4, a fixed injection rate of 5 Tg of $SO_2$ per year from 2020 to 2070. In both cases, the injection was at the Equator between 16 and 25 km in altitude, similar to how those same models would reproduce the Pinatubo eruption. As different models have varying

assumptions in terms of the initial Pinatubo plume, this protocol led to some discrepancies between models: a more in-depth discussion of this discrepancy between models can be found in Timmreck et al. (2018), which motivated the development of a modeling volcanic experiment named the Historical Eruption $SO_2$ Emission Assessment (HErSEA), and whose results have been recently described in Quaglia et al. (2022). Indeed, Pitari et al. (2014) reported large discrepancies in the amount of global aerosol optical depth necessary in G3 (i.e. 0.025 in GISS by 2070 and 0.1 in ULAQ-CCM) and a general disagreement over

the latitudinal distribution of the aerosols in the G4 experiment. The paucity of models that could reproduce the full cycle





from SO$_2$ oxidation to heterogeneous chemistry processes crucial for ozone chemistry was also highlighted. The termination of the implementation in 2070 has also produced interesting research over the impacts of a potential "termination effect," were geoengineering to be stopped abruptly (Robock et al., 2008; Jones et al., 2013; Parker and Irvine, 2018; Trisos et al., 2018).

In retrospect, these two experiments involved a learning process for GeoMIP. Experiment G3 was difficult to perform, as it involved regular radiative forcing calculations (often via a double radiation call), and many groups reported having to redo periods of the experiment because they injected too much or too little SO$_2$. Also, because the protocol specified that net top-of-atmosphere radiative forcing should remain at 2020 levels, temperature steadily increased in the simulations because the climate was already in imbalance in 2020. This exacerbated climate adjustments, increasing the spread and uncertainty in the
multi-model ensemble. As a result, few models participated in G3, and most of the analysis of this experiment was used to supplement analysis of G4 (Berdahl et al., 2014; Yu et al., 2015), although the runs were used, for instance, to study the impacts on agriculture (Singh et al., 2020), Atlantic hurricanes Moore et al. (2015), Arctic permafrost (Chen et al., 2020), and flood return frequency Wei et al. (2018).

In turn, one of the intended purposes of G4 was to "do the experiment the same way that [you] would simulate Pinatubo,"
and the groups would then look at the model spread in the climate outcomes. But because the models had such different micro-physics representations, aerosol distributions, and circulation patterns, it was difficult to attribute the spread to any particular processes or model features. However, these experiments did result in quite useful analysis and enabled some conclusions about tropical stratospheric aerosol geoengineering, amongst them: the potential stratospheric ozone depletion at high latitudes (Berdahl et al., 2014) and its effect; its capability to reduce (but not halt altogether) ice melting (Berdahl et al., 2014; Zhao
et al., 2017); and the dependency of the simulated precipitation reduction on the aerosol interaction with radiation (Ferraro and Griffiths, 2016). They were also used to study the impacts on ecosystems (Trisos et al., 2018). Nevertheless, they also provided important lessons about how to design a controlled experiment for GeoMIP and reinforced the community's notion that aerosol microphysics and circulation patterns are important contributors to model spread.

## 190  2.4    Compare and contrast: G6solar and G6sulfur

At the first GeoMIP meeting, there was a proposal (akin to what became the Testbed) for G3solar, in which the protocol for G3 was followed up using solar reduction instead of stratospheric sulfate aerosol injection. Although this proposed experiment did not receive much participation, the idea of comparing solar dimming and sulfate aerosols in identical protocols emerged as a Tier 1 experiment in CMIP6 in the form of experiments G6solar and G6sulfur (Kravitz et al., 2015). Similar to G3, the
amount of sulfate to be injected varies every year (or later modified to be every decade, due to the difficulty of calculating forcing in transient runs) to obtain a certain target: in this case, the aim was to reduce global mean surface air temperatures from those under an SSP5-8.5 scenario to those under a SSP2-4.5 scenario, "mitigating" the warming produced by high GHG concentrations. To do so, starting in 2020, models would reduce the solar constant (G6solar) or inject SO$_2$ in a band between 10°N and 10°S and 18 and 20 km in altitude (G6sulfur). The presence of two different experiments with similar targets allows





for a broader assessment of the differences between solar dimming as a proxy and an actual sulfate injection (Niemeier et al., 2013; Visioni et al., 2021a, and Fig. 2), and has allowed us to assess the contribution of stratospheric uncertainties to overall uncertainties in the climate response to geoengineering (Jones et al., 2021; Visioni et al., 2021b; Bednarz et al., 2022a). Of the 6 models that originally participated in G6, only two had comprehensive enough stratospheric chemistry to make them viable to assess ozone changes; at the same time, three used a prescribed aerosol distribution and three used actual $SO_2$ injections;

the overlap between those with comprehensive chemistry and those with interactive aerosols was two (CESM2 and UKESM) (Tilmes et al., 2022). Experiment G4 had a similar mix of explicit and prescribed representations of these processes.

In G6, modeling centers have demonstrated that it is feasible to modify the amount of intervention in the models even once per decade to maintain a pre-determined temperature target; the available comparison between temperatures under SSP2-4.5

(same global temperatures, different amount of $CO_2$) allows for a contrast throughout the entire simulation period of multiple scenarios and can also be used to understand when the emerging signal of a geoengineering deployment would be distinguishable from natural variability.

MPI-ESM performed two sets of G6 simulations with the same prescribed aerosol forcing but two different horizontal res-

olutions (roughly 200 km and 300 km, Muller et al. (2018)). A focused analysis of the differences in the climate response between these two versions could shed light on the impact of higher resolution modeling on projected geoengineering impacts.

There is an interesting observation that can be drawn from the G6 experiments and that multiple users of the data have noticed. Six years passed from when the experiment was officially proposed in 2015 (Kravitz et al., 2015), when the modeling

centers produced the simulations (early 2020 at the earliest) and when the first analyses came out in 2021 (Jones et al. (2021) with a subset of models; Visioni et al. (2021b) with all six). With geoengineering studies being a novel, fast-paced field, in those six years multiple discussions and studies have led many to question the relevance of both a high-emission scenario like SSP5-8.5 and the strategy aiming to "halve" warming to that of SSP2-4.5, as opposed to other, more moderate scenarios that try to discuss geoengineering in light of the Paris Agreement targets (see section 3.4 for instance); further, the injection strategy

of injecting in the tropical pipe has also been found to be sub-optimal Kravitz et al. (2019); Visioni et al. (2021a). In this sense, some may feel that analyses of G6 results may look outdated already due to the relevance of the specific scenario selected. While this is not necessarily true, as there is a great deal of merit in analyzing results from the latest iteration of CMIP6 models, it is a valid concern. If there is a lesson to learn it might be that there is a need for "future-proofing" the next generation of proposed GeoMIP experiments, so that they remain relevant for as long as possible.

## 2.5 Sea spray geoengineering: G3-SSCE, G4sea-salt and G4cdnc

As discussed in Section 2.2 above, GeoMIP developed several Tier 1 experiments to explore multi-model uncertainty in the climate response to marine cloud brightening. They were based on the G3 and G4 experiments described in Kravitz et al. (2013a), but substituting sulfur injections with the addition of sea salt.





The first one, G3-SSCE, was described in Alterskjaer et al. (2013); three models participated in that experiment, simulating
the effect of an increase of sea salt aerosols in the lower atmosphere. Later, the experiment G4cdnc prescribed increasing
the cloud droplet number concentration in all marine low clouds (lower in altitude than 680 hPa), which replicates the net
microphysical effect of MCB but without relying on different parameterizations of aerosol-cloud interactions. G4sea-salt on
the other hand involved direct aerosol injection into the marine boundary layer (between 30°S and 30°N), which is the most
realistic representation of MCB in GeoMIP but also runs the risk of resulting in a large model spread due to dependence on
aerosol-cloud interaction parameterizations as well as inter-model differences between cloud location and forcing strength.
Both experiments are somewhat less controlled than G1ocean-albedo, in that in G4cdnc and G4sea-salt, the MCB forcing will
only be applied where there are clouds, which differs between models.

Nevertheless, G4cdnc and G4sea-salt revealed important insights about MCB, including the importance of cloud differences
between models (Stjern et al., 2018) and that the aerosol direct forcing of sea salt aerosols can be quite important in MCB
applications (Ahlm et al., 2017). Arguably, more of the model spread in these MCB experiments was attributable to specific
processes than was the case for G3 and G4, which could reflect community investment in uncertainty: aerosol-cloud interac-
tions have received vastly more attention than stratospheric processes. Results from G4cdnc and G4sea-salt have only been
investigated in a few studies each (i.e. Xie et al. (2022)), so it is presently difficult to make conclusions about how one might
design a more controlled MCB experiment for Earth System Models.

## 2.6   Cirrus cloud thinning: G7cirrus

Due to the presence of some work on cirrus cloud thinning (CCT; Mitchell and Finnegan (2009); Storelvmo et al. (2013)),
GeoMIP proposed a Tier-1 experiment G7cirrus in which the fall speed of upper tropospheric ice crystals was increased (Muri
et al., 2014) to achieve a negative radiative forcing. However, due to large uncertainties in model representations of cirrus and
upper tropospheric ice water path, as well as a disconnect between the ice crystal distribution from increasing fall speed and
the distribution that would result from CCT (Gasparini et al., 2020), the GeoMIP community chose to re-classify G7cirrus as
a lower-tier experiment. This happened prior to most modeling groups conducting their simulations for Phase 6, which in part
explains why only two models have performed G7cirrus simulations to date.

Nevertheless, the tier of the experiment has historically not been a huge barrier to participation, so we suspect there are other
issues at play. Based on discussions with the community, there are three main reasons as to why G7cirrus has not received
much attention. From a scientific standpoint, during the first Gordon Conference on Climate Engineering held in 2017, some
members of the GeoMIP community pointed out that cirrus clouds and rather poorly represented in GCMs, and large challenges
remain even in the observational record about the main sources of cirrus clouds formations and their properties (Gasparini et al.,
2018; Sourdeval et al., 2018). Fundamentally, this would make the results of such an experiment rather untrustworthy.

From an operational point of view, G7cirrus is also not necessarily easy to perform, as it often requires code editing and
testing. We hypothesize about potential ways to mitigate this in Section 2.8 below. Despite not necessarily being a realistic
representation of CCT (as well as doubts about the effectiveness of CCT), a multi-model analyses of G7cirrus results would





still represent a learning opportunity for the community. Indeed, GeoMIP has thrived on learning from experiments that lack realism, most prominently G1.

### 2.7 GeoMIP6 Timeslice experiments

In addition to the Tier-1 experiments in GeoMIP6, there were several timeslice experiments proposed as Tier-2 simulations. These experiments involve 10-year simulations with fixed sea surface temperatures in which an external forcing is applied around a particular time, branched from the Tier-1 experiments. These were introduced to aid in separating the rapid adjustments from the feedback response, which was a major focus of GeoMIP analyses in previous iterations. As of the writing of this paper, few (if any) modeling groups have completed these experiments, and there does not appear to be widespread interest

in conducting them. We suspect this is for a few reasons. First, the GeoMIP community seems to be uncertain about the value of these timeslice experiments, so they have been deprioritized. Also, much of the analysis directions in GeoMIP have moved away from fast/slow response diagnostics, due perhaps to a stronger focus on a "gradual" deployment, obviating the need to complete the timeslice experiments.

While these experiments would still be useful, they were introduced at a time when the analysis they would engender was not

as popular. Perhaps the lesson learned is to design experiments that could serve multiple purposes, rather than a narrow purpose (diagnosing fast/slow responses). However, one could argue that lower tiers are well suited for this sort of specificity, and even if only a few models conduct those simulations, it would still be effort well spent. Nevertheless, the timeslice experiments proposed could easily fit within the spirit of other MIPs that are aimed at diagnosing these processes, indicating that it would be prudent to more actively pursue coordination between GeoMIP and other MIPs.

### 2.8 Past progress in official GeoMIP experiments: relevance in and outside the GeoMIP community

In our perspective, and based on our experiences and feedback from the broader community of researchers, GeoMIP has been a resoundingly successful MIP. New analyses and publications are continually underway, and attendance at annual meetings continues to increase. GeoMIP has emerged as a flagship activity in the geoengineering research community and has served as a common venue where people interested in this topic can interact, and new partners are always welcome.

The past experiments described in this section have been instrumental in highlighting high priority research areas for the community. While it has been pointed out that solar dimming has a limited value in representing stratospheric sulfate aerosol geoengineering (Visioni et al., 2021a), it is nevertheless a highly valuable experiment, since the straightforward setup allows us to be confident in the robust climate model responses to an experiment like G1 (Kravitz et al., 2021)). We are less confident in our understanding of stratospheric sulfate aerosols, and we are able to attribute those uncertainties largely to the complexities

of aerosol microphysical growth, aerosol distribution, and stratospheric circulation. In doing so, we have provided an evidence base for more targeted approaches, such as the GeoMIP Testbed experiments described in Section 3 below.

Nevertheless, as discussed earlier, participation has somewhat waned over the years. Fewer models are conducting the experiments, and the number of people leading GeoMIP papers has not kept pace with the growth of the GeoMIP community. Geoengineering research has suffered from a notable dearth of funding, so most people working on GeoMIP have volunteered





their time and effort. After doing that for 12 years, some may find it hard to keep up past levels of enthusiasm and time for involvement. At the same time, CMIP has moved to a "satellite MIP" approach, where each topic receives its own MIP (21 endorsed for CMIP6) with its own requirements for base simulations. Due to the controversial nature of geoengineering research, or simply that different groups have different interests, GeoMIP has not been universally highly prioritized by modeling centers for CMIP computer time. In addition, as discussed above, some of the GeoMIP experiments have had narrow purposes that may not align with individual researchers' interests, further reducing participation.

Nonetheless, GeoMIP experiments have been critical for some recent high-level reports such as: i) the 2022 Quadrennial Ozone Report by the WMO: without G6sulfur, there would have been no multi-model intercomparison of ESM results as related to the potential impacts of SAI on the ozone layer; ii) the IPCC AR6, GeoMIP-based papers contribute to an assessment of SRM in a cross-chapter box in Working Group I (CCB10) and a cross-working group in Working Group II (Chapter 16), iii) the National Academy of Science, Technology and Medicine report (of Sciences Engineering and Medicine, 2021) also extensively references GeoMIP and its related works.

We therefore think there are large opportunities to coordinate with other MIPs. While some may think that geoengineering itself does not overlap with the aims of other MIPs, the simulations and science objectives could serve dual purposes. As an example, during the design of CMIP6 experiments, we had conversations with the leads of the Cloud Forcing Model Intercomparison Project (CFMIP; Webb et al. (2017)); their interest in comparing the effects of solar and $CO_2$ forcing, as well as a delineation between fast and slow responses, align well with G1 and the timeslice experiments. Phase 6 of GeoMIP Kravitz et al. (2015) had an overshoot scenario that was loosely coordinated with ScenarioMIP (O'Neill et al., 2016) and led to further studies of overshoot scenarios in the geoengineering research community (Tilmes et al. (2020); also see Section 3.4). The 2022 WMO Quadrennial Report on Ozone, which required an assessment of possible changes due to SAI, has been an opportunity for renewed talks with the CCMI community, which resulted in the senD2-sai experiment (Table 1). Through more focused coordination efforts with other MIPs, we could co-design other experiments that serve multiple communities, thereby increasing the likelihood that those simulations are conducted and analyzed. This process could be aided if the World Climate Research Programme (WCRP) took a more active CMIP coordination role. Lastly, the analysis of output from geoengineering simulations has proven a viable and valuable means for involving researchers from countries that do not have the capacity for highly developed modelling programs, as has been demonstrated – albeit not to the extent as other modeling frameworks – through the DEGREES initiative (see Section 5.1.3 for further discussion of this initiative).

## 3    Current proposed testbed experiments and GeoMIP-adjacent experiments

Together with the experiments discussed in Section 2, numerous other experiments have been proposed and performed as Testbed experiments. Kravitz et al. (2015) initiated the GeoMIP Testbed whereby groups could propose simulations and conduct them with a limited set of models, providing a pathway toward formal adoption by GeoMIP if those simulations go well.





Here we discuss some of the proposed Testbed experiments, as well as other relevant geoengineering experiments that have been or could be leveraged by the community and replicated.

### 3.1 SO$_2$ and H$_2$SO$_4$ injections

Most model simulations of stratospheric aerosol injections have simulated the release of SO$_2$, due to the volcanic analogue. While this allows for some calibration based on observations of volcanic eruptions, we also know that using SO$_2$ injection results in large aerosols, which reduces the scattering efficiency, increases fall speed, and increases side effects (like stratospheric heating). Pierce et al. (2010) and following works (i.e., Benduhn et al. (2016); Vattioni et al. (2019)) have proposed direct injection of H$_2$SO$_4$ particles into the accumulation mode, which would avoid H$_2$SO$_4$ vapors formed from SO$_2$ oxidation from

coagulating on pre-existing particles and having them grow too much. A testbed experiment was carried out Weisenstein et al. (2021) comparing and contrasting the injection of SO$_2$ and H$_2$SO$_4$ with the aim of observing the response of a subset of GeoMIP models with interactive aerosol microphysics. The injection of 5, 10 and 25 Tg of S in either form was simulated in two different injection strategies, one uniformly spreading the aerosols between 30°N and 30°S at all longitudes, and one injecting at 30°N and 30°S in only one gridbox. As in G4, the injection of fixed amounts of materials highlighted models' differences in

their formation of aerosols, forcing efficiency and cascading impacts (such as the response of stratospheric dynamics, Franke et al. (2021)) and confirmed the possibility of H$_2$SO$_4$ injections as a way to constrain aerosol sizes towards more efficient radii. The use of three models with different aerosol treatments also allowed for more in-depth analyses of models' differences in terms of simulated size distribution, possibly also highlighting the need for more detailed aerosol microphysics in modal models.

### 3.2 G4SSA and CCMI-prescribed aerosol fields

The complex contribution of aerosol microphysics, chemistry and dynamical changes in determining the overall stratospheric response has been highlighted in many of the general GeoMIP experiments (Visioni et al., 2021b)). One way to constrain part of the response may be through prescribing an identical aerosol field in different models, in order to obtain a similar perturbation and understand how and why models differ in their projection of stratospheric heating and chemical changes to

an identical perturbation; such an experiment can also overcome the lack of a detailed interactive aerosol treatment in some ESMs. Tilmes et al. (2015) first proposed a similar experiment called G4 Specified Stratospheric Aerosols (G4SSA), offering the community a prescribed aerosol dataset determined using the ECHAM5-HAM microphysical model. At least one model with detailed chemistry but lacking a high top and stratospheric aerosol treatment performed the simulations as they were prescribed (Xia et al., 2017), and one model scaled the field to perform G6sulfur simulations (Visioni et al., 2021b). However, in

this latter case, while the model prescribed the aerosol distribution, it made its own assumption about the size distribution of the particles that form the optically thick cloud (Tilmes et al., 2022). This highlights the difficulties of properly performing such an experiment, making sure all models capture the same aerosol properties in order to reduce sources of uncertainty in projections.



Some insight about best practices for such an experiment could be gained by experiments with a similar philosophy described
in Zanchettin et al. (2022) for the Volcanic forcing MIP. After highlighting the inter-model disagreement in a Tambora-like
simulation in Clyne et al. (2021) where injection rates of $SO_2$ were prescribed, Zanchettin et al. (2022) discussed the prescrip-
tion of a volcanic forcing input between models, so as to focus on inter-model differences in the surface climate response. They
showed that, by combining the forcing prescription with a robust sampling of initial conditions between models (something
that is fundamental in volcanic simulations, but would not be in long term geoengineering ones), model disagreement could be
reduced and a further focus could be given to the climatic surface response.

The Chemistry-Climate Model Initiative (CCMI) has collaborated with some in the GeoMIP community to set up a new
shared experiment in order to support the new phase of CCMI (CCMI-2022) meant to inform upcoming WMO reports. In
particular, CCMI models expressed interest in an experiment similar to G4SSA in which Chemistry-Climate models could pre-
scribe the same aerosol distribution and observe changes in key stratospheric quantities. This experiment, senD2-sai, described
in the July 2021 SPARC newsletter (Plummer et al., 2021), will use a new aerosol field produced with CESM2-WACCM6,
symmetrical around the equator, maintaining SSTs fixed at present levels and running for 75 years from 2025 to 2100. The
synergy between GeoMIP and CCMI, with all the expertise such a joint effort could leverage, is an exciting opportunity for
GeoMIP.

### 3.3 Isolating the stratospheric heating contribution

The absorption of longwave radiation by the sulfate aerosols and the enhanced absorption of shortwave radiation by ozone
induced by aerosol scattering leads to a higher stratospheric warming that has long been known to produce numerous effects,
both in the stratosphere and in the troposphere. Different size, spatial distribution, persistence and chemical composition
of the injected aerosols would modify the specific of this warming. Studies of past volcanic eruptions (Robock and Mao,
1995; Polvani et al., 2019; Coupe and Robock, 2021; DallaSanta and Polvani, 2022) have revealed little consensus about the
dynamical effects of this stratospheric warming on surface climate. Furthermore observing systems are either sparse or, in the
case of satellite-based infrared sounders, influenced by the presence of the volcanic aerosols. Radiosonde and satellite-based
microwave sounders do indicate some warming, but the amount remains uncertain. So while experiments that prescribe the
aerosol field can highlight how different models warm the lower stratosphere differently, experiments that directly prescribe
a stratospheric heating perturbation (as in Simpson et al. (2019)) may help isolate model spread in the dynamical and surface
response, such as the models' response to El Ninos (Liu et al., 2022)). Nonetheless, these experiments could shed light on
which processes contribute more to the overall inter-model spread in experiments such as G6sulfur.

Two shared protocols have been proposed for this purpose: one aimed at observing surface changes induced by the strato-
spheric heating and one aimed at observing stratospheric dynamics changes. The fact that two different sets of models have
been used for this purpose highlights the possibility of exploring some of the processes involved in geoengineering through
the use of multiple tools: for surface changes, lower-resolution models with prescribed ozone and preindustrial conditions, that
allow for the production of larger ensemble of simulations in order to better quantify the significance of some of the observed





changes; for stratospheric changes, models with high stratospheric vertical resolution used for the QBOi experiments (Butchart et al., 2018).

One of the issues with determining a shared protocol for this kind of experiment is the determination of the amount of stratospheric warming to simulate. A higher warming signal obviously allows for a larger signal-to-noise ratio, but there might be questions over the realism or the relevance of such warming signals, particularly if that large heat flux results in nonlinearities in the model response. For instance, in the G6sulfur experiments, for a global increase in optical depth of 0.1, models provide a range of warming in the lower tropical stratosphere between 1 and 5 K to offset a surface warming of roughly

0.5 K, whereas by the end of the century the warming ranges between 5 and 14 K to offset a surface warming of (on average) 2 K. Simpson et al. (2019) and Visioni et al. (2021a)), who both imposed a stratospheric heating signal to isolate its impacts on surface climate, used an average value of 12 K Visioni et al. (2021a) that resulted from the GLENS simulations performed with CESM1(WACCM) offsetting almost 5 K of surface warming under an RCP8.5 scenario.

On the other hand, the first proposed protocols mentioned in this section decided to use a stratospheric heating perturbation

consistent with a forcing offset of 2 W/m$^2$ derived from Dai et al. (2018), which would result in a much lower average stratospheric heating. This choice will thus produce results that would be expected for a "moderate" deployment, and clearly a much lower signal for the surface perturbation obtained.

Both approaches have both merits and shortcomings, depending on their specific intent; nonetheless, both are a fundamental pieces of the puzzle in trying to separate uncertainties related to the dynamical perturbation that stratospheric sulfate would

produce, and that might be difficult to isolate in more comprehensive experiments such as G6sulfur. In this way, stratospheric heating experiments can be seen as complementary to experiments such as G1, that try to isolate the contribution of changes in the incoming SW radiation (albeit solar dimming experiments might also modify stratospheric dynamics, as pointed out in Bednarz et al. (2022a)).

### 3.4 Overshoot experiment

Tilmes et al. (2020) proposed a new GeoMIP testbed experiment that aimed to examine the response to multiple stratospheric aerosol geoengineering scenarios aimed at keeping temperatures at two targets (2°C and 1.5°C above preindustrial) not only under a high emission scenario (SSP5-8.5) but also under a scenario representing an "overshoot" (SSP5-3.4OS, Meinshausen et al. (2020)), that is, a scenario with very large carbon dioxide removal after 2040 which results in only a brief overshoot above the 2°C target. This idea of "peak shaving" has long been part of the discourse around geoengineering to represent temporary

deployment to keep temperatures down while mitigation and negative emissions efforts are ramped up (Long and Shepherd, 2014; MacMartin et al., 2018b). While this experiment could be part of a broader discussion around multiple scenarios involving high and low levels of intervention, SSP5-3.4OS is a Tier-2 experiment in ScenarioMIP, and thus modeling centers may not have the underlying emission scenarios simulations. Further, the geoengineering overshoot scenario uses the same feedback algorithm (Section 3.5 below) defined by Kravitz et al. (2017), which not many modeling centers yet have implemented.






Lastly, some modelers might feel that when it comes to simulating "policy-relevant" scenarios, one should prioritize those that appear to be more plausible (although while implausibility can be determined in some cases, plausibility is entirely subjective), and a scenario with very large amounts of greenhouse gas removal already by 2040 might be very well considered unrealistic. Meinshausen et al. (2020) describe SSP5-3.4OS as "geophysically interesting", and perhaps the proposed geoengi-
neering overshoot scenario should be viewed under the same lens. It also has the merit of fostering discussions over how to select "policy relevant" scenarios that might be important for future CMIP choices; for example, considering a geoengineering scenario that includes carbon removal could lead to studies that quantify the interaction between solar geoengineering and carbon removal holistically (Xu et al., 2020). In particular, as SSP5-3.4OS relies heavily on bioenergy crops to reduce the $CO_2$ concentration in the second half of the century (Melnikova et al., 2022), it will be interesting to investigate the interactions
between stratospheric aerosols and the potential for mitigation through BECCS.

### 3.5   Single point injection simulations and explicit feedback

The idea of using control theory to modify the annual $SO_2$ injection amount (MacMartin et al., 2018a; Kravitz et al., 2016, 2017) has been gaining traction in the geoengineering research community. In recent years, multiple experiments using CESM in various configurations have used a feedback algorithm to demonstrate that surface impacts of stratospheric aerosol geoengineering
can be reduced if, instead of simulating equatorial or tropical injections of $SO_2$, these injections are distributed over other latitudes (namely, 15°N, 15°S, 30°N and 30°S) to control not just global mean temperatures, but also inter-hemispheric and equator-to-pole temperature gradients (see Kravitz et al. (2017); Tilmes et al. (2018); Richter et al. (2022). The merits of this strategy have been discussed in depth by Kravitz et al. (2019) as they compare to equatorial injections, leading many to ask if future GeoMIP experiments should incorporate an explicit strategy dimension (MacMartin et al., 2022); for example, future
GeoMIP experiments could include off-equatorial injection strategies instead of the equatorial strategy used in G4 and G6sulfur.

Two obstacles limit this option: the working of the feedback algorithm that directly imposes in the model how much $SO_2$ to inject every year may depend on exactly how each climate model imposes emissions in the model, requiring both an interactive $SO_2$-to-sulfate aerosol treatment, which not all models have, and some software engineering applied to each model to make the
algorithm work. Secondly, developing the control algorithm that manages more than one degree of freedom required previous sensitivity studies that determine the response of single-point injections at the required injection locations, as done by Tilmes et al. (2017) and MacMartin et al. (2017) for CESM1(WACCM). A GeoMIP testbed experiment reproducing the single-point injection and developing a similar algorithm for multiple models has been performed and described by Visioni et al. (2022) and Bednarz et al. (2022b), but these studies were built upon the substantial body of work that had already been conducted.
Any additional models wishing to engage with such a simulation would need to conduct their own single injection locations, which is computationally expensive. The feasibility of expanding this idea to a larger host of GeoMIP models still needs to be discussed, and some groups may opt to study simpler strategies that still reduce impacts compared to equatorial injection but are easier to implement.





The ARISE-SAI (Assessing Responses and Impacts of Solar climate intervention on the Earth system with stratospheric aerosol injection) protocols could offer an example of how such a feedback-driven experiment could work: the idea behind the underlying scenario choices have been described in MacMartin et al. (2022), while Richter et al. (2022) clearly lays out the injection protocols, the overall set-up for one model (CESM2-WACCM), and the required variables needed for analyses (that for things like aerosol fields might expand on what is recommended by CMIP in terms of required variables). A similar

protocol (or the ARISE protocol itself) could be adopted as a future GeoMIP experiment.

### 3.6 G4foam

Experiment G4Foam (Gabriel et al., 2017) involves surface brightening, similar to G1ocean-albedo (Section 2.2), but only in selected oceanic regions where the forcing is expected to be amplified via cloud feedbacks. This idea is related to the idea of Green's function approaches to forcing that have been gaining traction recently in the study of climate feedbacks (Dong et al.,

2019). Indeed, Harrop et al. (2018) showed that in CESM there are certain oceanic regions where, if one adds a positive heat flux, global mean temperature decreases. This raises the question as to whether there are high leverage locations where small amounts of forcing could have robust, disproportionate effects. This area requires substantial further study, as it is not presently known whether these results are replicable in other models and, if so, what the physical mechanisms behind these effects may be. This experiment also reinforces the idea that issues central to geoengineering are also central to climate science in general

– in this case, the relationships between forcing, rapid adjustments, feedbacks, and response.

## 4 Future experiments

Based on the review of current and past experiments, as well as community discussions, we have formulated opinions about what future experiments could be considered in geoengineering research. In the last two GeoMIP meetings, the GeoMIP community broadly agreed that the lack of person-power, the computational expense, and the general uncertainty over future

directions of CMIP indicated that the proposal of new Tier-1 experiments could be premature (Visioni and Robock, 2022); this of course might change in 2023, spurred by recent analyses, new community input, a notably increased interest in the theme by climate and impact scientists, and, perhaps, this piece. However, there was also a general agreement that we could use this time to better evaluate the models we currently have through process-based, more narrowly defined experiments to better constrain the inter-model spread observed in current GeoMIP iterations (Visioni et al., 2021b). These process-based experiments often

do not need to be run with fully-coupled Earth System Models, relieving part of the computational cost and making these experiments interesting to a wider community. These efforts need to be run in parallel with efforts aimed at defining future policy-relevant scenarios that include geoengineering (Tilmes et al., 2020; MacMartin et al., 2022). In this section we describe some of these ideas and the reasons why we believe they are high priority for the research community.

Unlike other modeling experiments for which measurements are readily available to constrain model uncertainty (like vol-

canic eruptions or past stratospheric ozone evolution), there has been no implementation of geoengineering on a climate-relevant scale, and thus there are no observations of geoengineering. However, diagnosing the spread in the response of dif-



ferent models in a widening set of impacts on the natural and man-made world is always useful, and may inform future plans for research (MacMartin and Kravitz, 2019) and, perhaps, contribute to the discussion around the need for future outdoor experiments (Golja et al., 2021).

## 4.1 G6polar

In addition to equatorial injection, Robock et al. (2008) tested the climate effects of injecting $SO_2$ into the polar stratosphere (67.5°N). Its inefficiency at reducing global mean surface temperatures and its impact on global precipitation, predominantly shifting the Intertropical Convergence Zone (ITCZ) was noted (Haywood et al., 2013). More recently, Lee et al. (2021) re-evaluated polar injection by considering seasonal injections in the spring which would result in less perturbation overall, and highlighted the efficacy of such a strategy in restoring sea ice and the benefit of having to loft material at lower altitudes given the lower height of the tropopause there (Smith et al., 2022). This illustrates that geoengineering may have important trade-offs that need to be uncovered and discussed to inform future decisions around whether and how geoengineering might be deployed.

The need to address this question opens up the opportunity of devising a testbed experiment aimed at analyzing the response to Arctic stratospheric injection in multiple models. However, because ITCZ shifts are a known consequence of single-hemisphere injection, care should be taken in considering the precipitation response, perhaps by devising an experiment that injects in both the North and South polar stratosphere in order to balance the forcing. If such an experiment is to be carried out, care should be also taken over the determination of a target: Lee et al. (2021) prescribed a fixed injection rate of 6 Tg-$SO_2$ in various seasons and simply observed the resulting changes (like in a G4-like experiment), whereas Lee et al. (2022) and Jackson et al. (2015) devised a target based on restoration of sea ice. It is likely that a fixed injection rate would be easier to analyze in the beginning (and would ultimately inform development of a feedback algorithm to target specific objectives), leading to clearer differences between different models. A well-designed experiment such as G6polar could also serve as a useful tool to better understand the potential of SAI to provide an emergency brake on some high-latitudinal tipping elements of the Earth system (Lenton et al., 2008), which have been understudied up to know in the context of geoengineering.

## 4.2 Specified dynamics

The effects of inter-model differences in large-scale circulation have been highlighted in multiple venues, especially in CCMI (e.g., Eichinger et al. (2019)). For stratospheric aerosol geoengineering, Niemeier et al. (2020) showed large differences in the baseline residual vertical velocities, aerosol confinement in the tropical pipe, and the surface response to an aerosol perturbation. Large scale differences in the circulation result in different latitudinal distributions of the aerosols, especially if injections happen close to the tropical pipe (Laakso et al., 2017; Bednarz et al., 2022b), but aerosol microphysical growth and the dynamical perturbation itself also play important roles (Visioni et al., 2022). It could therefore be posited that another way to separate specific differences between models can be through the nudging of stratospheric dynamics to common values, as was done in CCMI in the RefC1-SD experiments (Orbe et al., 2020). Nudging climate models to an actual meteorology has also been done for simulations of volcanic eruptions, for instance by Schmidt et al. (2018), resulting in better agreement with observations in





terms of the global mean forcing produced. However, Chrysanthou et al. (2019) observed that in CCMI simulations, nudging chemistry-climate models did not constrain long-term trends in the residual vertical upwelling in the ensemble compared to free-running simulations.

So while there could be merit in observing the response of GeoMIP models to nudged simulations in terms of the aerosol dis-
tribution, care should be taken in designing those simulations in multiple models. Another way could be the use of Chemistry-transport models (CTMs) like GeosChem that do not have interactive circulation at all, which allows for an actual prescription of circulation patterns. However, CTMs seldom have detailed aerosol microphysics in the stratosphere compared to some CCMs (Visioni et al., 2018).

### 4.3 Sensitivity to aerosol parametrization

While technically not a GeoMIP experiment, Laakso et al. (2022) explored using the same climate model (ECHAM-HAMMOZ) but with two different aerosol microphysical models. This highlighted the sensitivity of the baseline climate and the response to sulfate injections to the aerosol model, underscoring the need to pay more attention to aerosol parametrizations. Similarly, Visioni et al. (2022) compared two versions of GISS ModelE2.1, one with a bulk aerosol treatment and one with a quasi-modal microphysics, finding even larger discrepancies. One possible way to more systematically understand differences between mi-
crophysical schemes could be a protocol analogous to Weisenstein et al. (2007), where an intercomparison was performed between a zero-dimensional box model, a 2-D model (the Atmospheric and Environmental Research (AER) sectional model with multiple configurations), and a 3-D model (the Global Modeling Initiative (GMI) 3-D chemical-transport model). A similar comparison between models used for GeoMIP simulations with sulfur injections could highlight potential areas of needed improvements, and perhaps even define a "minimal standard" of complexity (for some experiments) that climate models should
have before they can be considered to produce robust results.

### 4.4 Radiative transfer

Recent studies have shown that some inter-model discrepancies in the response to $CO_2$ can be traced to the shortwave radiation code (Chung and Soden, 2015; DeAngelis et al., 2015). Differences in radiative transfer have long been recognized, for example, in the Radiative Transfer Model Intercomparison Project (RTMIP; Collins et al. (2006)). Boucher et al. (1998) performed
a comparison of the shortwave response to sulfate aerosols for 15 radiative transfer codes produced by 12 different groups under a wide range of specified size distributions and aerosol and atmospheric optical properties. Unlike for the response to greenhouse gases, where the inter-model spread ranged up to 40%, Boucher et al. (1998) found that the standard deviation of normalized forcing near the optimum size for maximum scattering was small (8%). This highlights the importance of investigating fundamental model processes – we do not want to conclude erroneously that the response to geoengineering is uncertain
if the actual uncertainty is in the radiative transfer code and thus somewhat independent of the forcing mechanism.





It would be extremely valuable to perform a similar intercomparison with current radiative transfer codes, especially those used or whose usage is planned for GeoMIP studies. This could be done for both shortwave and longwave forcing; in both cases, GeoMIP models show large spreads in efficiency of the forcing and in lower stratospheric warmings Visioni et al. (2021b); Tilmes et al. (2022). Understanding the behavior of the radiative codes separately could help understand which areas of model improvement require more focus. Recent proposed methods, such as by Jones et al. (2017) involving comparing a very specific set of diagnostics with a line-by-line radiative model, could suggest a way forward. This method has been adopted by the Radiative Forcing Model Intercomparison Project-Aerosol Instantaneous Radiative Forcing Component (RFMIP Aerosol-IRF) (Pincus et al., 2016), and a collaboration between the two MIPs could be extremely informative. Such focused experiments, including analyses of the radiative kernels, could help better understand and constrain the global response to solar geoengineering on the hydrological cycle Kravitz et al. (2013b, 2021); Tilmes et al. (2013); Kleidon et al. (2015) as has been done for a $CO_2$ increase (DeAngelis et al., 2015).

### 4.5 How do we devise the next experiments for Marine Cloud Brightening?

Simulations in MCB can, to some degree, be separated into two categories. First, can clouds be brightened, and if so, by how much and under what conditions? Second, assuming clouds can be brightened, what are the climate effects of brightening clouds in specific areas? Experiment G4cdnc began addressing the second one, brightening all clouds where they existed. Indeed, this second category is well within the wheelhouse of Earth System Models, as they are adept at understanding the climate response to imposed forcing. Specific experiments could be designed to test MCB in a multi-model capacity, such as choosing certain locations, different-sized regions, or seasons, and performing Green's function approaches (analogous to those discussed in Section 2.2 above) in each model to exert a specified amount of forcing. One could envision either using the model's internal cloud field or prescribing a cloud field; each approach has its advantages and disadvantages.

The first category, as to whether clouds can be brightened, is somewhat more complicated. We argue that this category of studies is outside of the auspices of GeoMIP, although coordination with GeoMIP would certainly be valuable. Understanding this question deals with fundamental questions in aerosol-cloud interactions, namely the susceptibility of cloud albedo to aerosol changes. This has been and continues to be the largest source of uncertainty in climate science (Chen et al., 2021). The answer varies depending upon location, atmospheric conditions, and the background aerosol state Lee et al. (2016), among numerous other factors. Addressing these questions in a model context could be done through a series of large eddy simulation (LES) experiments. The advantage of LES is that because of the small grid size (sometimes on the order of meters), many cloud processes are resolved instead of parameterized, which relieves a substantial source of uncertainty. LES experiments could be conducted under a wide variety of atmospheric conditions and aerosol injection strategies to understand under what conditions marine low clouds can be brightened and by how much. This in turn could be used to constrain further GeoMIP experiments, in which regions to be brightened are selected dynamically rather than assuming MCB automatically works. This would allow the research community to establish an upper bound on the effectiveness of MCB, which will directly inform future deployment decisions.



## 4.6 Impacts assessment

A long-established need in geoengineering research is to understand downstream impacts, such as agriculture, ecosystems, and food/water security (Irvine et al., 2017). GeoMIP has been used in the past as driving information for impacts models (e.g., Xia et al. (2014)). Nevertheless, this process at present is woefully insufficient for quantifying benefits and risks of geoengineering. Any uncertainties or spread in the Earth System Models (as discussed above) will propagate through impact models, which have their own uncertainties, leading to a wide range of results, even independent of different future scenarios of climate change and geoengineering. There is some effort to perform intercomparisons of impacts models, such as in the InterSectoral Impacts Model Intercomparison Project (ISI-MIP; Rosenzweig et al. (2017)) or the Agriculture Model Intercomparison Project (AgMIP; Rosenzweig et al. (2012)), to understand and reduce uncertainties in impacts assessments. Coordination between those MIPs and GeoMIP is in its initial stages, but developing a path forward, including specific GeoMIP experiments that are designed for impact assessment, would be of benefit to a wide variety of communities. A similar argument could be made for the ecological impacts of geoengineering, which are poorly understood (Zarnetske et al., 2021). It is essential to mention the DEGREES Initiative in this respect (https://www.degrees.ngo). One of the fundamental point of DEGREES is that researchers in the global South, who are experts in climate impacts of most concern to their region, may also have access to high-frequency (at least daily) geoengineering model output in order to properly assess specific impacts that may be related to e.g. agriculture, or floods, or loss of the cryospheric elements of the system. The great advantage of bringing in these often poorly-funded groups is that the research is community-driven from the grassroots upwards. Impacts that may interest the modeling community of GeoMIP may be quite different from those of most impact locally. The scope for impact studies might be expanded in future to e.g. saline intrusion into ecosystems, fisheries, pests and diseases, human health, heat stress and interactions with tropospheric pollutants.

## 4.7 Summary of outstanding scientific questions

Sections 3 and 4 offer many examples of questions that the GeoMIP community is interested in. This might feel like a scatter-shot of very different ideas, and in some ways it is so: the topic the GeoMIP community tries to tackle is a vast problem where the single pieces are always interconnected. Here, we highlight some of the key questions that continue to motivate our work.

**What is the consensus of the most up-to-date models of the likely effects of solar geoengineering on the Earth System?** is certainly the one at the core of GeoMIP itself: we use highly advanced and, hopefully, independent climate models and try to discern commonalities in their response to various geoengineering techniques. If agreement is high for a certain aspect of the system, we can presume that that's the most likely response that the real Earth system would have. In general, the topic of "tipping points" Lenton et al. (2008) and the potential of some form of solar geoengineering to delay or alt the emergence of dangerous instabilities in the planetary system is also one that deserves far more attention in the future.

Highlighting disagreement, however, may be as important as highlighting consensus: **Why do models disagree in their response to solar geoengineering?** This may look similar to the previous question, but it shifts the focus from the projection





itself to the tools used to derive it: is the multi-model average of the response "closer to the truth" than any single model or are there individual models that are "better at" capturing the effects of geoengineering than others (and how do we even
characterize "closer" and "better" in a highly complex system)? If so, how can we constrain models with observations (past, and future) to narrow down the uncertainty from multi-model ensemble simulations? In its latest Assessment Report, the IPCC highlights in multiple places the "large uncertainties" related to geoengineering. Together with a less vague quantification of such uncertainties, GeoMIP can also help identify areas of necessary model agreement.

Moving to the various techniques tested, questions will differ based on the level of advancement in the field: for Marine
Cloud Brightening and Cirrus Cloud Thinning, a major question that remains is **Could these geoengineering techniques measurably affect the global climate?** For Stratospheric Aerosol Injection, which has been studied more and for which more solid evidence behind its natural analogue, volcanoes, exist, questions might be focused more on some of the details of the implementation: **What are the impacts of different strategies of implementations of SAI?** is also a very broad question, but one that focuses on the potential design aspect of geoengineering rather than on the uncertainty aspect of it, even if the two are
connected (it is pointless to think about designing something of which we know very little about).

## 5   The role of GeoMIP

GeoMIP is a framework for the intercomparison of geoengineering experiments using climate models. GeoMIP does not include all geoengineering research, and not all geoengineering research should go through GeoMIP: this is a fundamental point to put forward in order to understand the present and future role of GeoMIP itself. There are clear examples of successful mod-
eling efforts outside of GeoMIP, with the Geoengineering Large Ensemble (GLENS, Tilmes et al. (2018)) being one of the most prominent. GLENS focused on producing a large ensemble (20 members) of simulations using one model (CESM1(WACCM)) and one, at the time, novel strategy (using an automated feedback loop capable of determining where and how much $SO_2$ injections should go in the next year). This allowed for a more thorough exploration of, for instance, signal-to-noise emergence MacMartin et al. (2019) and extreme events (Aswathy et al., 2015; Pinto et al., 2020; Tye et al., 2022). Both GeoMIP and
GLENS have enabled numerous studies through the DEGREES Initiative, enabling developing countries to assess changes they may experience under geoengineering.

With the urgency of climate change increasing impacts on societies and ecosystems, there is a great need to continue and accelerate geoengineering research (of Sciences Engineering and Medicine, 2021), and a large merit in its scientific exploration through the use of coordinated multi-model experiments. We see two primary reasons for this. The first, primarily scientific, is
that the overreliance on the results of just one climate model can be a potential pitfall and result in overconfidence in results that are model-specific. Multi-model comparisons can highlight outliers (not necessarily wrong, but different as compared to the mean) which can in turn spur further research behind the reasons why: this is for instance the case for the higher climate sensitivity of some CMIP6 models ((Zelinka et al., 2020). This in turn can direct effort toward improving our understanding of model performance, with experiments such as those described in Section 4. The second, that can be mainly defined
as "social", is that knowledge-sharing and expertise-building are both at the core of GeoMIP and of fundamental importance



in the field of geoengineering studies. In the previous sections we have tried to outline how much the process behind defining new experiments for GeoMIP relies on "learning by doing (and making mistakes)". Yearly in-person GeoMIP meetings (except two online during the COVID pandemic) have been an opportunity for the community to grow, and learn, and share knowledge across different continents and levels of expertise: for each meeting, every year, a report is available (see the list at http://climate.envsci.rutgers.edu/GeoMIP/publications.html) detailing both the discussions that took place and the novel science presented by researchers at every stage of their career.

Indeed this opinion piece was strongly inspired by the discussions had during the Gordon Conference on Climate Engineering that took place in Newry, ME in the Summer of 2022, both in terms of the future plans and in term of the level of enthusiasm we saw in the meeting, which involved over 50 early career researchers (while the first Gordon Conference, held in 2017, had fewer than 10). The yearly GeoMIP meeting was held during one of the afternoons of the Gordon Conference both times, and a very large portion of the people that were at the Conference attended. This social aspect of GeoMIP is as valuable as the purely scientific one and, to some degree, those two aspects are inseparable.

With these reflections in mind, in this section we want to look even more broadly at the future of GeoMIP and discuss what direction we think the community should (and should not) move towards.

## 5.1  What shouldn't be in GeoMIP's purview?

Here we want to summarize some of the most recent research or discussions around future geoengineering directions, but that we think are either too premature to be considered in GeoMIP or simply do not fall in its purview.

### 5.1.1  Is it necessary to simulate pathological scenarios in GeoMIP?

In the past, there have been studies aimed at understanding "pathological" SRM deployments or events: the effect of an abrupt termination (Jones et al., 2013; Trisos et al., 2018) or those of a single-hemispheric deployment (Haywood et al., 2013; Jones et al., 2017). The point of analyzing such scenarios is mostly to answer the question "what could go wrong?", which is extremely important. However, it could also be misinterpreted in communication of the results as "this is what SRM would (always) do" (Reynolds, 2022), and therefore there can be a legitimate discussion over whether such scenarios are needed. In particular, because GeoMIP has served as a community-building and organizing effort, the project speaks with a loud voice; if GeoMIP explores pathological scenarios, that may send the signal that such scenarios are considered to potentially be realistic.

At the time the termination effect was proposed for inclusion in GeoMIP, few studies had explored it (e.g., Wigley (2006); Trisos et al. (2018)). As such, there were numerous questions about the detailed effects of termination, including whether models would respond on similar timescales and with similar accelerations of climate change. The community has since learned a great deal from those simulations, and while there may remain open questions about termination, we presently see no need to conduct such investigations in numerous models. A similar discussion could be had for single-hemispheric deployment. This scenario is interesting and instructive from a basic response standpoint and has resulted in fundamental knowledge that has informed geoengineering discussions. There is also some value in comparing the results of these experiments in multiple



models (Haywood et al., 2016) to understand the degree of changes and whether there are robust signals. However, it is hard
to envision a situation in which geoengineering would only be deployed in a single hemisphere for multiple decades. As such,
these experiments may not be appropriate to adopt as formal GeoMIP experiments.

There has recently been increased discussion of uncoordinated geoengineering or "rogue actors" where there is no central-
ized decision making about an optimal aerosol distribution or target, but rather multiple actors may be attempting to meet
multiple, independent, perhaps conflicting targets Rabitz (2016). The lack of climate model simulations of this scenario in the
geoengineering literature has been discussed before (McLaren and Corry, 2021). Nevertheless, we argue that this is not an
experiment that should be adopted by GeoMIP. Policy experts or game theory analysts could in principle devise a scenario
involving multiple actors. But the challenge would then be translating such a scenario into a specific experiment that could be
run consistently in multiple models. Moreover, it is not obvious that there would be advantages to conducting such a specific
simulation; for example, GLENS involved independent injections at four different locations, and it is unclear how different a
simulation with decentralized geoengineering would be from GLENS. While we do not rule out the possibility of this being
explored in the future, especially as more concrete policy decisions emerge over the coming years, but we do not see a role for
GeoMIP in these scenarios at this time.

### 710   5.1.2   Alternate Materials for Stratospheric Aerosol Injections

To date, all GeoMIP experiments involving solar geoengineering through the injection of aerosols in the stratosphere have
consisted of injections of $SO_2$ or $H_2SO_4$ (Weisenstein et al. (2021); Section 3.1). Global modeling of alternate aerosols is
sparse, with some exceptions for black carbon or titania (Kravitz et al., 2012; Ferraro et al., 2015; Jones et al., 2016). Alter-
nate, potentially less impactful materials such as calcium carbonate have been proposed to overcome the problem of ozone
changes and lower stratospheric temperature perturbations (see for instance Keith et al. (2016); Dykema et al. (2016)), but
since those materials are not naturally occurring in the atmosphere, information is still being sought on the chemical reaction
rates (Dai et al., 2020). For this reason, it is clearly premature to consider such experiments for GeoMIP, but the potential
for future intercomparisons would arise if the inclusion of these novel materials were to be tested in multiple climate models
independently.

### 720   5.1.3   Subgrid-scale processes

One of the shortcomings of using climate models for SAI simulations is the inability to resolve fine-scale behaviors once the
materials have been injected. In Section 2.5 we discussed some of the cloud processes that are unresolved at the scale of Earth
System Model grids, requiring parameterization. For stratospheric aerosol injection, there are additional uncertainties at the
sub-grid scale. For example, models often assume that once $SO_2$ is injected into a grid box it becomes immediately uniformly
distributed. This is uncharacteristic of diffusion processes or mixing forced by an aircraft jet. Additionally, many relevant





weather and climate processes happen at the sub-grid scale, which is particularly relevant for quantifying changes to extreme temperature and precipitation.

While many of these topics are certainly relevant for GeoMIP and may affect model capabilities and experiments in the future, as an Earth System Model intercomparison, studying the importance of subgrid-scale processes is outside of the purview

of GeoMIP and is better left to individual studies. There are numerous other MIPs and perturbed parameter ensembles that are focused on understanding parametric and structural uncertainty; through coordination and communication, such lessons can be conveyed to GeoMIP without requiring a specific GeoMIP experiment. New work involving a Lagrangian plume model that can be embedded in ESM grid boxes (Sun et al., 2022) is underway, but it needs further testing and has not been widely implemented, so using this capability in GeoMIP is premature.

Dynamical downscaling, either using a regional model or a regionally refined global model, is a standard technique for gaining finer-scale information. Downscaling has been shown to add value to GCM data over regions of topography, at land-water boundaries and in better representing extremes. Currently these methods are being tested for geoengineering applications (Wang et al., 2022) so adoption by GeoMIP is still premature. However, future efforts to coordinate between GeoMIP and the Coordinated Regional Climate Downscaling Experiment (CORDEX; Kotlarski et al. (2014)) could prove to be fruitful in

coming years, perhaps in concert with efforts to increase impact assessment.

## 5.2  What should GeoMIP do next?

As an official MIP under CMIP, the primary goal of GeoMIP for the coming years should be to prepare for CMIP7, as well as continuing to harvest potentially valuable results from the efforts put into the CMIP6 simulations. This is especially important considering the potential for results from GeoMIP to be included in future international reports like the Intergovernmental

Panel on Climate Change (IPCC) and World Meteorological Organization (WMO) reports, as it has been in the past).

The experiments proposed in Section 4 would go a long way towards building more trust in the models used for future projections, in improving the modeling tools at our disposal in future iterations, and towards better understanding the underlying processes that compose the overall response of the Earth system to geoengineering. However, in IPCC reports there is a strong push towards the use of complex emission scenarios, developed with the use of Integrated Assessment Models (IAMs) that can

represent a range of possible and plausible futures (Nakicenovic et al., 2000) spanning greenhouse gasses emissions, land use, and population changes. The process that led to the development of the set of scenarios used in the Assessment Report 6 of the IPCC has been described in detail by O'Neill et al. (2016). The whole process was defined by the ScenarioMIP Scientific Steering Committee, which included extensive discussions with members of multiple scientific communities, numerous MIPs and IPCC task forces. The main objectives that guided the decisions were: i) facilitating integrated research between climate

analyses, scenario analyses and feedbacks between climate and society; ii) addressing targeted science questions regarding particular components of the overall forcing; iii) better quantifying projection uncertainties based on multi-model ensembles.

On the other hand, and with no surprise considering the smaller scope, previous GeoMIP scenarios have been defined by a much smaller (and narrower in terms of expertise) community, mostly taking into account the modelers' needs for geoengineering scenarios that were straightforward to implement in different climate models. The amount of person-time available





to produce the GeoMIP simulations is perhaps two orders of magnitude lower than that made available by modeling centers for ScenarioMIP, and therefore it is unsurprising that GeoMIP has made use of available SSPs and added geoengineering on top. This is likely to continue to be the case for some time, but nonetheless the geoengineering research community should explore how geoengineering might be incorporated directly into the scenario development process. Learning from other MIPs and the history of geoengineering, we think it is important that future CMIP7 scenarios include geoengineering as well (in all

its various forms). In our view, the scenarios that include them should address the following criteria:

1. **Plausibility**: in terms of possible start date, amount of cooling, characteristics of the deployment and more, the scenarios should reflect to the best of our current knowledge realistic deployment options.

2. **Policy relevance**: the scenarios should be capable of informing policy-makers with regards to the possible outcomes of a geoengineering deployment by considering more than one possible scenario and strategy, as the analyses of just

one case may lead to confusing a scenario-specific result with a result that is applicable to all geoengineering scenarios. Scientific relevance: the analyses of the developed scenarios should aid in our scientific understanding of the outcomes of a geoengineering climate.

3. **Scientific relevance**: the analyses of the developed scenarios should aid in our scientific understanding of the outcomes of a geoengineering climate.

4. **Reproducibility**: the requests to the modeling teams should be as simple as possible in order to ensure the participation of as many models as possible, minimizing possible errors and ensuring reproducibility.

These four criteria present some tension between them: for instance, a scenario like G1 is very scientifically relevant, has high signal-to-noise ratio, and is easily reproducible, but not plausible nor policy relevant, whereas a scenario like G6sulfur might be scientifically and policy relevant but is not plausible (the start date has passed; current emissions do not seem to track

the SSP5-8.5 scenario), and future scenarios like those described in MacMartin et al. (2022) might not be easily reproducible in many climate models, which makes them not presently relevant for GeoMIP. In general, more plausible scenarios will have lower signal-to-noise; however, there is the question of whether an effect (both direct and indirect) that is too small to be detected in a plausible simulation matters (To whom? And for what?) and needs to be investigated. We view this tension as a good thing, and indeed scenarios should not strive to meet all four criteria equally (although meeting the reproducibility criteria

should be considered essential for inclusion as a Tier-1 GeoMIP experiment). Our purpose is not to prescribe at this stage what future GeoMIP experiments should be. Rather, we argue that proposed experiments should be interrogated to ensure that they are being true to purpose so that experiments are widely adopted by modeling groups and their results can be appropriately communicated and are useful for the scientific and policy-making community.

Moreover, the process for coming up with scenarios needs to be explicit about its intended audience. In particular, new

scenarios should carefully consider: i) the need for an inclusive process that takes into account multiple lines of expertise across multiple fields: climate science, ecosystem sciences, social sciences; ii) the needs of the community of modelers on which GeoMIP depends; iii) the needs of the community of scientists that want to use GeoMIP output. Balancing these three





different sets of needs out will also result in some tensions (for instance, between the need for high-frequency output required by the impact modeling community and the difficulty of storing the large amount of data that would be produced), but it is a

necessary discussion both in order to ensure that the scenarios are as representative as possible, and in order to make sure that as many researchers as possible feel represented in GeoMIP.

To offer some specific examples of the needs of the users of GeoMIP, there is a growing community of ecologists interested in understanding the impacts of geoengineering on ecosystems (Zarnetske et al., 2021) that have already successfully used GeoMIP simulations. On the other hand GeoMIP data has only been used in 3 out of the 12 papers published to date in the

DEGREES project, funded by the DECIMALS fund, which aims to supports teams of researchers in developing countries (https://www.degrees.ngo/publications/, last accessed September 5th, 2022). In some cases this disparity makes sense – for example, DEGREES teams who want to study extreme events (e.g., Abiodun et al. (2021)) would benefit from using a large ensemble like GLENS to obtain a good signal-to-noise ratio or be able to sample rare events. For some other groups, as the number of models increases, accessibility and usability of the data becomes harder, especially for teams with poor internet con-

nectivity or teams that require computing power to conduct their analyses. GLENS is a single model, and all data is available in one place, which can make it easier to use. There are important advances being undertaken for CMIP analyses, particularly by the Pangeo community (Odaka et al., 2020) who have enabled analysis of CMIP6 data in the cloud without requiring users to download terabytes of output. Nevertheless, widespread use of GeoMIP output, including data accessibility, is currently an unresolved issue.


Lastly, DEGREES teams have repeatedly reported their need to focus on small scale, regional impact assessments, which often requires downscaling results from available climate models. Although several teams have conducted regional impact modeling analyses (e.g. hydrology, heath, ocean modeling), only one study has employed dynamical downscaling of GeoMIP results, a study using WRF to downscale G4 output over northeast China in comparison with statistical downscaling (Wang

et al., 2022) and another to statistically downscale G4 output over the Indonesian Maritime Continent (Kuswanto et al., 2022). To statistically downscale climate data on a regional scale, however, the underlying data need to reliably replicate the climatological features of the baseline climate, which in some cases some models might fail to do. The issue of "which models are best to use" is therefore also a problem which DEGREES teams are faced with.

## 6   Conclusions

The Geoengineering Model Intercomparison Project community has been active for more than 10 years, and its participants span numerous countries on multiple continents (Visioni and Robock, 2022). Here we have reviewed past and present proposed GeoMIP experiments, including those determined by the community as high priority (Tier-1) and those proposed by specific members as "Testbed experiments", and have reflected on the potential future development of GeoMIP.

While not a review of the state-of-the-art understanding of SRM, which exists elsewhere (e.g., Kravitz and MacMartin

(2020)), critically assessing all available GeoMIP experiments is a useful exercise for understanding how the field of geoengi-





neering modeling studies has evolved, as it gives an idea of the current areas where research has focused, and how to move forward from there.

The inclusion of recent experiments and of numerous new potential experiments, or areas where experiments could be devised, has multiple aims, many of which have been brought up repeatedly in publications and at various meetings of the research community. Amongst them: i) to offer the community a way to devise new, more specific analyses on current experiments that might have been underutilized and could still be leveraged; ii) to serve as a starting point for the community to more carefully devise new experiments, both more specific ones as we detailed, and also for future CMIP experiments to be included in international assessments; iii) lastly, currently many calls are available for "more research" into solar geoengineering, such as the report from the National Academy of Science (of Sciences Engineering and Medicine, 2021). Nevertheless, those calls for "more experiments" or "more research" often lack detail, which is a barrier to action. Based on the discussions amongst the large GeoMIP community over the years, in this piece we have provided a critical examination of where GeoMIP's research has led and what gaps need to be filled, and we leave with a number of conclusions which could potentially inform future research direction and aligned programs.

One obvious gap, which has been repeatedly highlighted by the community, is a lack of people-time more than simply computer time, both in defining shared protocols for experiments and in analyzing the available output. As a simple example, even just determining injections of $SO_2$ in exactly the same grid-box in multiple models requires more work than just specifying a certain height in the protocols: different models may have different kinds of vertical coordinates and different ways to specify exogenous emissions. Coordination between modeling centers and providing attention to these details is therefore a crucial part of a successful multi-model comparison. Related to this is a lack of funding to support geoengineering research, in terms of both people and computer time, and a lack of capacity in the Global South to operate in this space. As southern countries are typically the most vulnerable to climate change, entraining local scientists into geoengineering research is especially important (Rahman et al., 2018).

We should underscore that multi-model comparisons are definitely not the only way to understand, constrain, and eventually reduce model uncertainty. As for climate change, uncertainty does not come from a single source. In our context, as explained for instance by Hawkins and Sutton (2009), it can come from internal variability, uncertainty in the climate response, scenario uncertainty, and parametric or structural uncertainty. Internal variability can be better studied in the context of large ensembles of simulations with single models, of which some are available for geoengineering studies already (Tilmes et al., 2018; Richter et al., 2022)): the use of multiple realizations from very similar initial conditions allows for a better separation of a given forcing signal from noise derived from the inherent chaoticity of the atmospheric and oceanic system, and makes exploring the timing of the emergence of such a signal easier (MacMartin et al., 2019; Tye et al., 2022). GeoMIP is suited to explore uncertainties in the climate response as it allows an exploration of structural differences between models to a standardized forcing, but it does not directly address single-model uncertainty based on specific parameters in the physical representation of various aspects of the climate system: for such an endeavor, perturbed-physics ensembles within a single model may offer





a much clearer answer. One example of such a perturbed-physics ensemble is the proposed Pinatubo Emulation in Multiple models (PoEMs) experiment in the Interactive Stratospheric Aerosols MIP (ISA-MIP) (Timmreck et al., 2018), in which modeling teams already simulating the 1991 Pinatubo eruption are asked to perform additional simulations modifying some of the possible parameters (in their case, such as aerosol nucleation, coagulation or sedimentation rates) in order to span the

possible space of parameters that most closely matches observations of the eruption. Some of the GeoMIP experiments, and especially some of the experiments we have proposed in Section 4, may help single modeling teams better understand where to focus their efforts in terms of parameters to analyze, and may even leverage existing protocols like PoEMs does in order to longitudinally compare both against other realizations with the same model, and with other models. An initial effort along these lines was explored by Irvine et al. (2014) for G1 using HadCM3. Carefully designed protocols that can incorporate both

model and parametric uncertainty would provide numerous advantages for quantifying and attributing uncertainty in processes.

Scenario uncertainty, which over multi-decadal scales is larger than other sources in the CMIP context (Lehner et al., 2020), can also be very hard to sample in the GeoMIP context. Moreover, scenario exploration has historically required multiple decades of simulation for each scenario, which limits the number of scenarios that can be explored. Considering multiple,

but shorter, simulations could perhaps free up the resources needed to span the scenario dimension, but at the cost of perhaps under-sampling the long term climatic response: for future iterations of experiments that are part of CMIP, the community will have to make some choices on this aspect, or perhaps distinctly consider short-term process details and long-term response characteristics in separate experiments. A possibility could be to also focus on studies that try to develop better emulators that are applicable in the geoengineering context: some currently exist, and may work better for some variables than for others

(MacMartin and Kravitz, 2016). These allow researchers to analyze multiple scenarios a posteriori after a fewer number of possibly higher signal-to-noise scenarios has been simulated with the full host of climate models, and ultimately verify the emulator with a subset of available climate models afterwards.

There continues to be an important role for GeoMIP in geoengineering research, both scientifically and as a community ef-

fort. We recommend that for activities in GeoMIP, participants carefully evaluate their purpose and participation to ensure that the objectives of GeoMIP continue to be met, as well as supporting the ongoing development of GeoMIP. We also recommend active coordination with other MIPs, World Climate Research Program activities like the Stratosphere-troposphere Processes And their Role in Climate (SPARC) and the Safe Landings Lighthouse activity, and other geoengineering research efforts to identify synergies that will increase the use of GeoMIP efforts and encourage more participation in GeoMIP. Issues and un-

certainties in geoengineering are rarely exclusive to that field – rather, they are often common to climate science research in general. Bringing geoengineering research into the mainstream, perhaps in part through efforts made by GeoMIP, will benefit both geoengineering research and broader climate research efforts.



*Data availability.* No code or data was produced for this work

*Author contributions.* D.V. and B.K. wrote the piece, with substantial contributions from all other coauthors.

*Competing interests.* The authors have the following competing interests: One or more co-authors are in the editorial board of ACP

*Acknowledgements.* The authors would like to deeply thank the many scientists that, over the years, have contributed their time and effort to GeoMIP. Support for B.K. was provided in part by the National Science Foundation through agreement SES-1754740, the Indiana University Environmental Resilience Institute, and the Prepared for Environmental Change Grand Challenge initiative. The Pacific Northwest National Laboratory is operated for the US Department of Energy by Battelle Memorial Institute under contract DE-AC05-76RL01830. A. R. is supported by U.S. National Science Foundation grants AGS-2017113 and ENG-2028541. S.W. was supported by JSPS KAKENHI Grant Number JP2103668.



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
