# Peer review of "Opinion: The Scientific and Community-Building Roles of the Geoengineering Model Intercomparison Project (GeoMIP) - Past, Present, and Future"

_Atmospheric Chemistry and Physics, 2022_

## Author Comment (AC1)

**Response to Reviewer 1**

Reviewer comments are in black, authors' responses are in blue

The GeoMIP project has produce a lot of useful results so it is good to take a step back and think about lessons learned.

The review of past experiments and pointers to some of the resulting papers is very useful and interesting.

We thank Ken Caldeira for his comments. We have tried to include all of his suggestions in the revised manuscript.

**Major notes:**

This paper would be substantially improved with the addition of a section titled something like "Lessons learned". If you were to start this project over again knowing what you know now, what would you have done differently? [Since there might be multiple perspectives on what should have been done differently, this might be an opportunity to share several perspectives. In this reviewers perspective, this lessons learned section would be the most important section of this paper.

Similarly, it might be good to pull together what I would see as a section on cross cutting issues. The two issues that I see as being raised at several points are:

1. To what extent should simulations attempt to be "realistic" and to what extent should the simulations be highly stylized aimed at facilitating more straightforward analysis?
2. Where should the balance be between simulations that may in some sense be "better", but be difficult for modeling groups to perform, versus simulations that might not be as useful, but might be easier for modeling centers to perform?

There are probably similar cross-cutting questions that you might want to address, for example: When is it enough to have a small number of groups do a simulation and when do you really need a large number of groups to do a simulation? How do you draw a balance between then number of simulations that people need to perform versus the number of ensemble members for each simulation? How to think about the GEOMIP demands on people's time versus everything else they need to be doing?

I would not expect to see resolution on all of these questions, but maybe a couple of sentences showing the thinking on all of these questions might be useful.

**Some of this material is already in Section 5.2 and the Conclusions section. Nevertheless, I think adding a "Lessons learned" section would be highly useful, and a "Cross-cutting issues" might be a place to focus discussion on some of the questions raised in Section 5.2 and the Conclusions.**

**The reviewers can accept or reject my "cross-cutting issues" suggestion but I hope would adopt my "Lessons Learned" proposal would be adopted.**

We thank the reviewer for his suggestions. We feel like many of the topics he is discussing were already there, so in trying to avoid lengthening the piece too much, we have restructured the manuscript moving many of the opinions around past experiments from the end of section 2 and other sections to a new section with the title suggested. We have also tried to highlight more in the Conclusions the "cross-cutting issues" he mentioned.

**Minor notes:**

**[line] comment**

**[24] Eliminate word "these" (stylistic)** Done

**[43-45] Provide citation for IS92a claim.** We have provided a citation to Pedersen et al. (2020): their figure 1 clearly shows how close IS92a has been to mean growth rates during the 1990-2019 period.

**[67] Properly capitalize of project name.** We followed the capitalization present in all of their reports (see i.e. https://cordis.europa.eu/project/id/226567/reporting) which never capitalized other words except the first. We now added ' ' to highlight the name, however.

**Table 1. Maybe make another column for the background scenario.** Thank you for the suggestion, we have added that and cleaned up the table.

**Figure 1. It might be helpful in this figure or in an additional figure to make it clear which CMIP scenario is the reference "ungeoengineered" case relating to each geoengineering case. At the very least this could be in the figure caption.** We have tried different options but that clutters the figure too much. Given that the reference scenarios are now more explicit in Table 1, we have added a reference to the table.

**Figure 2: Expand figure caption to explain all labeled points in the figure. For what years are this? Is it really the standard deviation so low, or are these perhaps standard errors? If the values for G6Solar, G6sulfur, are compared against SSP2-4.5 values, might it be a good idea to show the SSP2-4.5 value on the figure? Do something to let people know which geoengineering case is related to which case without geoengineering.** Thank you for your suggestions, we have tried to make the figure

clearer. We cannot include SSP2-4.5 values because they are the reference values, but we have tried to specify more things in the revised figure and caption.

[Figure]

New caption: *A comparison of global temperature (K) and precipitation (%) changes for some Tier 1 GeoMIP experiments across CMIP5 and CMIP6. Points represent the multi-model averages for each experiment, shaded areas represent 2 multi-model standard errors. Values for G1 and 4xCO2 (CMIP5, 13 models averaged) and G1ext and 4xCO2 (7 models) are from Kravitz et al. (2021), comparing against piControl values in the last 40 years of the experiment (years 11-50). Values for G6solar, G6sulfur and SSP5-8.5 (6 models) are from Visioni et al. (2021b), comparing against SSP2-4.5 values in the last 20 years of the experiment (2080-2099)*

**[104-105] Please mention whether the reduction was the same in each model or different to achieve a temperature balance.** We added some clarifications, and this phrase: "*For instance, such a comparison showed that between the models that performed*

*this experiment across two generations, the value of solar reduction needed ranged from 3.80 to 5.00 % (Kravitz et al., 2021).*"

**[124] Please mention whether the reduction was the same in each model or different to achieve a temperature balance.** Mentioned, as above.

**[221-226] Some discussion of the use of SSP2-4.5 as a reference state rather than SSP5-8.5 would be appreciated. Was the choice to be more "realistic" worth having a higher signal-to-noise ratio? From the discussion on these lines, it seems researchers wanted to be more "realistic", but maybe it is better to hit models with a hammer to see how they behave with more extreme forcing.** We have highlighted this trade-off in an additional phrase that reflects this point: "*Perhaps excessive focus on "realism" -whatever the current opinion on that is at a given moment- is good for communicating results and convincing modeling teams to consider performing a set of simulations, but might result in scenarios that perhaps do not hold the test of time as well as simpler, higher signal-to-noise experiments like G1.*"

**[228-229] This discussion of "future proofing" might be expanded and discussed later along with the above questions. Is the goal to be "realistic" or to understand how models behave? How are these competing goals best balanced?** Noted, refer to largest change.

**[Section 3 and 4] For each of these subsections, it might be good to start each section with the main scientific question that each project is intended to address. (For example, line 509 mentions a question in a section that has no questions in it.)** Good point. To find a better balance between length and descriptiveness, we have decided to expand the title of each subsection to make sure it reflects which question it is supposed to answer.

**[619] Something akin to this boldfaced question should appear in each of the subsections of Sections 3 and 4. (Maybe not a bad idea to do this for Section 2 also.)** See above.

**[777-788] These kinds of questions about tradeoffs in design should get more prominence.** We wholeheartedly agree!

---

## Author Comment (AC2)

**Response to Reviewer 2**

Reviewer comments are in black, authors' responses are in blue

**Overview:**

**This manuscript covers GeoMIP past and present. This includes discussion of successful and non-successful experiments and addresses what has been learned from GeoMIP. It finally addresses recommendations on how GeoMIP should progress in the future. It is categorized as an opinion piece. However, it's more of a review than an opinion piece. The title notes scientific and community-building roles of the project, but the paper is really just a review of past GeoMIP experiments, suggestions for future experiments, and doesn't really describe how there has been community building. The paper is also long and a big disjointed and rambles at times. I'd suggest deleting section 5.1 to shorten things a bit as well.**

We thank the reviewer for their comments. While the piece is relatively long and does review the experimental design and results obtained by GeoMIP, we don't believe that we could achieve what we have in this piece with a review article. Review articles do not allow the mixing of academic review, reflection and opinion that an opinion piece does. While this opinion piece is long, we believe that this is necessary given the breadth and scope of GeoMIP.

Based on your comments and those of the other reviewer, we have restructured the piece, to more clearly bring out the lessons learned and cross-cutting issues that were distributed throughout the sections in the initial submission. We feel that this revised version is significantly improved and addresses the reviewer's concerns about the structure and flow of the piece.

**Comments delineated by section**

**Section 2 starts off talking about different Tiers. Why are some high priority and others lower priority? Describe for the reader not familiar with MIPs what the difference is between Tier 1, 2 and 3 (or however many there are).** We feel like the current introduction explains the Tiers enough, but we have further specified that Tier 1 experiments are higher priority.

**and 2.1-2.8 all sound like they are "past" experiments...you should be clear about what you mean by "present" or delete it from the section title.** We have specified in the text that we meant past as in CMIP5, and present as in CMIP6.

**Section 2.8 belongs more in a summary than in the middle of the paper.** Based on the suggestions from reviewer 1 as well, we have moved this to a different sectionfurther down in the manuscript.

Section 3 says it is current proposed testbed experiments....are they current or proposed? Some of the text seems to indicate they are proposed, but in other cases it sounds like they experiments have already occurred. How about making the section title just Testbed experiments and GeoMIP adjacent experiments (although I'm not sure what "GeoMIP adjacent" actually means. Thanks for the suggestion. We have changed the title to "*Testbed experiments and other relevant experiments*".

Section 4 is titled "Future experiments" But it seems that section 3 was talking about proposed experiments (which seem like they'd be in the future). It seems this section is really talking about potential experiments to look at processes, or maybe it's better to call them Future testbed experiments.

See above.

Section 4.7....this should be a different topic (not under Future experiments). And, perhaps, it belongs more as a subset of section 5 or in the Conclusions.

We have moved this to the new section 5 together with the old section 2.8.

A few more substantive scientific comments:

line 348 says" possibly also highlighting the need for more detailed aerosol microphysics in modal models. " Does it perhaps suggest use of microphysical models that aren't model...perhaps sectional models as well? We meant to use the word "climate" models, and not "modal", here. Which yes, suggests the need for sectional aerosol microphysics (as we discuss later on in Section 4.3)

paragraph line 534-538: seems like there should be some mention that running a CTM or nudging (or replaying) to a common transport does not allow simulation of any transport changes that are caused by the aerosol heating, or any strat-trop interactions that may occur, so then is not fully simulating surface climate or strat ozone impacts. Thank you, this is a good point. We have added a phrase explaining this. "*In general, the use of such methods would prevent the simulation of the actual interactions between the stratospheric heating produced and the large-scale circulation: while this would not simulate the full response occurring in the atmosphere, it may help in separating the dynamical feedback from the uncertainties resulting from the aerosols.*"

Line 574 -576 says " Simulations in MCB can, to some degree, be separated into two categories. First, can clouds be brightened, and if so, by how much and under what conditions? Second, assuming clouds can be brightened, what are the climate effects of brightening clouds in specific areas? " The simulations are not separated into 2 categories, but the questions regarding MCB are separated into 2 categories. Corrected.

**There also needs to be some more thought as to whether GCM comparison runs are really useful for assessing the viability of MCB. They don't resolve the key processes, so it all boils down to parameterizations. I am not sure you can say (as in line 593/594) that any of these simulations will "directly inform deployment decisions". I actually suggest reading https://www.pnas.org/doi/full/10.1073/pnas.2118379119** Thanks, good point. We have amended the phrase to state this more clearly "*This would allow the research community to establish an upper bound on the effectiveness of MCB, and to produce better assessments of the potential large-scale circulation response, which would be a necessary part of an overall assessment of MCB research (Diamond et al., 2022)*"

**line 530: you may also want to consider this paper (https://gmd.copernicus.org/articles/13/717/2020/) in regards to potential issues with specified dynamics runs.** Added, thank you!

**Line 652: says " With the urgency of climate change increasing impacts on societies and ecosystems, there is a great need to continue and accelerate geoengineering research (of Sciences Engineering and Medicine, 2021), and a large merit in its scientific exploration through the use of coordinated multi-model experiments". I would suggest also noting that there is need for model verification using observations, and in particular identifying the key processes that models need to represent for various climate intervention techniques. Therefore, included within GeoMIP could be a coordinated model/measurement intercomparison. You don't just need to highlight outliers, but need to assess accuracy.** The reviewer raises an important point. While we are reluctant to claim a need for field campaigns dedicated to studying climate intervention (as this is a fraught issue and is handled much better in other sources, like the NASEM report), we do agree that there could be model/measurement comparisons as part of basic climate science or opportunistic measurements from geoengineering analogues (like volcanic eruptions). We have attempted to clarify this point in the revised manuscript.

**Editorial type comments/suggestions/questions:**

**line 22: This sentence "The comparison of results from nominally identical experiments in multiple, distinct climate models can be a very useful tool for understanding models' biases, robustness in the climate response to external forcings, and for partitioning sources of uncertainties in future climate projections Lehner et al. (2020)."**

**"comparison of results" is not a tool. Suggest a rewrite to "The comparison of results from nominally identical experiments in multiple, distinct climate models is useful for understanding models' biases, for assessing robustness in the climate response to external forcings, and for partitioning sources of uncertainties in future climate projections (Lehner et al., 2020)."** Thank you, we have reworded as suggested.

**general editing comment:  in many cases the references within the text are not properly formatted.  For example, in the sentence noted above, Lehner et al. (2020) should be written as (Lerhner et al., 2020)** Noted, we have tried to fix the issues.

**Line 32:  why do you call these " satellite MIPs"?** We have changed this (as suggested below) to endorsed MIPs elsewhere, and removed the "satellite" word from this line directly"

**Line 29-38 could be deleted.  It doesn't add to this paper at all.** As part of the aim of this work was to offer an introduction to this topic to students, we feel like this gives good context.

**Line 44: change "have, so far, been largely insufficient," to "have been insufficent"** Changed.
**line 51: Recognizing what facts?  Suggest a rewrite to "Because the goal of 1.5 or 2 degrees warming  seems unobtainable, around 10 years ago, an international group of researchers (Kravitz et al., 2011) proposed a new framework to coordinate climate modeling experiments to study proposals for solar geoengineering (also known as Solar Radiation Modification or Climate Intervention), aimed at understanding the impacts of proposed methods to offset the warming produced by an increase in greenhouse gases by directly intervening in the Earth's radiative balance."** We have rewritten this phrase to be clearer, thank you.

**line 55: change "targeting" to "increasing"** Changed.

**line 56/57:  delete " (for a comprehensive review of the scientific aspects raised by geoengineering techniques, see for instance"** Thank you for the suggestion. Again, as we hope for this paper to provide an introduction to the topic for students, we find it helpful to be explicit in pointing out that there are other references more focused on the scientific aspects of the impacts we discuss here.

**line 73, since CCT is a common abbreviation, change cirrus thinning to cirrus cloud thinning** Done.

**line 137:  I'd suggest you emphasize simulating in a multi-global model context has proven challenging (since there higher resolution models do a better job on MCB processes).** Done, thank you.

**line 140:  You might also mention that one of the reasons that this is difficult (or nigh on impossible) in a global model is that the key microphysical processes can only be parameterized in a global model.** Good point. We added this.

**line 261 change " cirrus clouds and rather poorly" to " cirrus clouds are rather poorly"** Changed.

line 301, use some other term besides "satellite MIP" which is somewhat confusing because that term does not appear on the CMIP web page (nor in any google search). CMIP refers to them as CMIP6-Endorsed MIPs.  you could change to "endorsed MIP approach" to match the CMIP terminology. Changed.

line 323/324 says " This process could be aided if the World Climate Research Programme (WCRP) took a more active CMIP coordination role."  What exactly do you suggest WCRP do? In our re-shuffling of some of the sections, we have removed this phrase.

paragrapn from line 313 to 327:  Give some actual suggestions of co-designed experiments.  Rather than starting with "We think there are ...." start with "Possible coordination experiments with other MIPS include..." We have added a call to Section 4, where we discuss some more concrete experiments.

line 336/337 says " we also know that using SO2 injection results in large aerosols"  How do you know this?  Is this a model or measurement result? We have changed this to "modeling studies indicate".

line 373 says " support the new phase of CCMI (CCMI-2022) meant to inform upcoming WMO reports. "  You might as well say to inform the 2026 WMO/UNEP Scientific Assessments of Ozone Depletion. Added, thank you!

610-613 " The scope for impact studies might be expanded in future to e.g. saline intrusion into ecosystems, fisheries, pests and diseases, human health, heat stress and interactions with tropospheric pollutants. "  don't say "to e.g."  just delete the e. G. Fixed

line 887:  SPARC is not an activity, it is a WCRP core project, and you might want to also consider collaborations with the ESMO core project (Earth System Modelling and Observations) We have fixed this, and thank you for the suggestion, which we have added.